# LINE-1 protein localization and functional dynamics during the cell cycle

**Paolo Mita[1]\*, Aleksandra Wudzinska[1], Xiaoji Sun[1], Joshua Andrade[2], Shruti Nayak[2], David J Kahler[3], Sana Badri[4], John LaCava[1,5], Beatrix Ueberheide[1,2], Chi Y Yun[3], David Fenyö[1], Jef D Boeke[1]\***

[1]Institute of Systems Genetics (ISG), Department of Biochemistry and Molecular Pharmacology, NYU Langone Health, New York, United States; [2]Proteomics laboratory, NYU Langone Health, New York, United States; [3]High Throughput Biology (HTB) Laboratory, NYU Langone Health, New York, United States; [4]Department of Pathology, NYU Langone Health, New York, United States; [5]Laboratory of Cellular and Structural Biology, The Rockefeller University, New York, United States

**Abstract** LINE-1/L1 retrotransposon sequences comprise 17% of the human genome. Among the many classes of mobile genetic elements, L1 is the only autonomous retrotransposon that still drives human genomic plasticity today. Through its co-evolution with the human genome, L1 has intertwined itself with host cell biology. However, a clear understanding of L1's lifecycle and the processes involved in restricting its insertion and intragenomic spread remains elusive. Here we identify modes of L1 proteins' entrance into the nucleus, a necessary step for L1 proliferation. Using functional, biochemical, and imaging approaches, we also show a clear cell cycle bias for L1 retrotransposition that peaks during the S phase. Our observations provide a basis for novel interpretations about the nature of nuclear and cytoplasmic L1 ribonucleoproteins (RNPs) and the potential role of DNA replication in L1 retrotransposition.
DOI: https://doi.org/10.7554/eLife.30058.001

\*For correspondence:
paolo.mita@nyumc.org (PM);
jef.boeke@nyumc.org (JDB)

**Competing interests:** The authors declare that no competing interests exist.

## Introduction

Retrotransposons are genetic elements that move within the host genome through a 'copy and paste' process utilizing an RNA intermediate. 17% of the human genome is made up of copies of the Long Interspersed Nuclear Element-1 (LINE-1 or L1) retrotransposon (*Lander et al., 2001*). L1 is the only autonomous human retrotransposon able to 'jump' through a process called retrotransposition. The full length, active L1 retrotransposons consists of a 5' untranslated region (UTR) containing a bidirectional promoter (*Speek, 2001*; *Swergold, 1990*), two open reading frames (ORFs), ORF1 and ORF2, separated by a short inter-ORF region and a 3' UTR with a weak polyadenylation signal (*Doucet et al., 2015*; *Dombroski et al., 1991*; *Burns and Boeke, 2012*; *Burns, 2017*). A short, primate specific, third ORF of unknown function called ORF0 was also described in the 5'UTR of L1 with an antisense orientation compared to ORF1 and ORF2 (*Denli et al., 2015*). Upon transcription by RNA polymerase II, the 6 kb bicistronic L1 mRNA is poly-adenylated and exported into the cytoplasm. LINE-1 exhibits 'cis-preference' (*Wei et al., 2001*), the mechanism of which is unknown. A popular model posits that newly translated ORF1 and/or ORF2 proteins preferentially bind the mRNA molecule that encoded them (*Boeke, 1997*). Interestingly, translation of the second ORF, ORF2p, is mediated and regulated by an unconventional process that remains poorly understood (*Alisch et al., 2006*).

**eLife digest** Only two percent of our genetic material or genome are occupied by genes, while between 60-70 percent are made up of hundreds of thousands of copies of very similar DNA sequences. These repetitive sequences evolved from genetic elements called transposons.

Transposons are often referred to as 'jumping genes', as they can randomly move within the genome and thereby create dangerous mutations that may lead to cancer or other genetic diseases. LINE-1 is the only remaining active transposon in humans, and it expands by copying and pasting itself to new locations via a process called 'retrotransposition'. To do so, it is first transcribed into RNA – the molecules that help to make proteins – and then converted back into identical DNA sequences.

Previous research has shown that LINE-1 can form complexes with a series of proteins, including the two encoded by LINE-1 RNA itself: ORF1p and ORF2p. The LINE-1 complexes can enter the nucleus of the cell and insert a new copy of LINE-1 into the genome. However, until now it was not known how they do this. To investigate this further, Mita et al. used human cancer cells grown in the lab and tracked LINE-1 during the different stages of the cell cycle.

The results showed that LINE-1 enters the nucleus as the cell starts to divide and the membrane of the nucleus breaks down. The LINE-1 complexes are then retained in the nucleus while the membrane of the nucleus reforms. Later, as the cell duplicates its genetic material, LINE-1 starts to copy and paste itself. Mita et al., together with another group of researchers, also found that during this process, only LINE-1 RNA and ORF2p were found in the nucleus.

This shows that the cell cycle dictates both where the LINE-1 complexes gather and when LINE-1 is active. A next step will be to further investigate how the 'copy and paste' mechanisms of LINE-1 and the two LINE-1 proteins are regulated during the cell cycle. In future, this may help to identify LINE-1's role in processes like aging or in diseases such as cancer.

DOI: https://doi.org/10.7554/eLife.30058.002

ORF1p is an RNA binding protein with chaperone activity (*Martin and Bushman, 2001*; *Khazina et al., 2011*) while ORF2p contains domains with endonuclease (EN) and reverse transcriptase (RT) activity (*Feng et al., 1996*; *Mathias et al., 1991*; *Weichenrieder et al., 2004*). It is believed that ribonucleoprotein particles (RNPs), consisting of many ORF1p molecules, as few as one or two ORF2ps and one L1 mRNA (*Khazina et al., 2011*; *Basame et al., 2006*; *Dai et al., 2014*) form in the cytoplasm. The RNP is then imported into the nucleus through a still uncharacterized process. L1 RNPs accumulate in cytoplasmic stress granules (SGs) and processing bodies (PBs) (*Goodier et al., 2007*; *Hu et al., 2015*; *Doucet et al., 2010*), but the role of these cellular structures in the L1 life-cycle is still controversial. L1 RNPs accumulate mainly in the cytoplasm but nuclear ORF1p was also observed in a certain percentage of cells using cell lines and cancer specimens (*Sokolowski et al., 2013*; *Sharma et al., 2016*; *Doucet et al., 2016*; *Goodier et al., 2007*; *Harris et al., 2010*). While several specific antibodies against ORF1p have been raised (*Taylor et al., 2013*; *Wylie et al., 2016*; *Doucet-O'Hare et al., 2015*), a highly effective antibody against ORF2 protein that would allow definitive observation of protein localization is still lacking. Antibodies against LINE-1 ORF2p have been recently developed (*De Luca et al., 2016*; *Sokolowski et al., 2014*) but the much lower amount of expressed ORF2p compared to ORF1p makes the study of ORF2p expression and localization difficult. To overcome these difficulties, tagged ORF2ps have been employed (*Taylor et al., 2013*; *Doucet et al., 2010*).

In the nucleus, L1 endonuclease nicks the DNA at A/T rich consensus target sites (5'-TTTT/AA-3') (*Feng et al., 1996*) and through a process called TPRT (Target Primed Reverse Transcription) (*Cost et al., 2002*; *Luan et al., 1993*) inserts a DNA copy into the new genomic target locus. During TPRT, ORF2p EN domain nicks the DNA and the newly formed 3'OH end is then used by the RT domain of ORF2p to prime the synthesis of a complementary DNA using the L1 mRNA as template. A second strand of cDNA is then synthetized and joined to adjacent genomic DNA.

L1 lifecycle is extensively entwined with host cellular processes. Several proteins and cellar pathways have been shown to restrict or support L1 retrotransposition and life cycle (*Goodier, 2016*). RNA metabolism (*Belancio et al., 2008*; *Dai et al., 2012*), DNA damage response (*Servant et al.,*

2017) and autophagy (*Guo et al., 2014*) are a few cellular processes shown to affect LINE-1 retro-transposition. Progression through the cell cycle was shown by several groups to promote L1 retro-transposition (*Shi et al., 2007*; *Xie et al., 2013*) but a molecular understanding of this aspect has been elusive. Because of the importance of the cell cycle in efficient retrotransposition, it has been proposed that, as for some exogenous retroviruses (*Goff, 2007*; *Suzuki and Craigie, 2007*), nuclear breakdown during mitosis could represent an opportunity for entrance of L1 into the nucleus (*Xie et al., 2013*; *Shi et al., 2007*). This hypothesis, never directly tested previously, was challenged by studies demonstrating effective retrotransposition in non-dividing and terminally differentiated cells (*Kubo et al., 2006*; *Macia et al., 2017*).

To shed light on the role of the cell cycle on different aspects of L1 life cycle we explored the nuclear localization of L1 proteins and the retrotransposition efficiency of L1 in different stages of the cell cycle in rapidly dividing cancer cells. Here, we use imaging, genetic and biochemical approaches to show that in these cells, the L1 lifecycle is intimately coordinated with the cell cycle. LINE-1 encoded proteins enter the nucleus during mitosis and retrotransposition appears to occur mainly during S phase.

## Results

### Analysis and quantification of ORF1p and ORF2p expression and cellular localization

Previous works (*Doucet et al., 2016*; *Luo et al., 2016*; *Sokolowski et al., 2013*; *Goodier et al., 2007*; *Goodier et al., 2004*; *Sharma et al., 2016*; *Rodić et al., 2014*; *Taylor et al., 2013*; *Branciforte and Martin, 1994*) have shown varying localization of ORF1 protein in cells growing in culture or in mammalian specimens from various organs and tumors. In particular, the nuclear localization of LINE-1 encoded proteins has been sparsely studied and the mechanisms driving import of L1 retrotransposition intermediates into the nucleus are largely unknown.

We therefore set out to characterize the cellular localization of L1 ORF1p and ORF2p in human cells overexpressing recoded (ORFeus) or non-recoded L1 (L1rp) with a 3xFlag tag on the ORF2p C-terminus (*Taylor et al., 2013*). Immuno-fluorescence staining of HeLa-M2 cells (*Hampf and Gossen, 2007*) overexpressing ORFeus, showed clear expression of ORF1p and ORF2p (*Figure 1A and D* and *Figure 1—figure supplements 1*,*2*, *Videos 1* and *2*). As previously observed (*Taylor et al., 2013*), ORFeus ORF1p was detected in virtually all the cells ($\cong$ 97%) whereas ORF2p, encoded by the same bicistronic L1 mRNA also expressing ORF1p, was detected in just a subset of cells ($\cong$10% using a rabbit anti-Flag antibody and $\cong$20% using a more sensitive mouse anti-Flag antibody) (*Figure 1B*, bar graph and inset). This pattern of expression is most likely due to an unknown mechanism controlling ORF2p translation (*Alisch et al., 2006*). Interestingly, when non-recoded L1 was over-expressed, only 44% of cells displayed ORF1p expression. Overexpression of ORFeus-Hs and L1rp with or without the L1 5' untranslated region (5' UTR) excluded the possibility that the reduced expression of L1rp compared to ORFeus was due to the presence of the 5' UTR (*Figure 1—figure supplement 3*) (*Chen et al., 2012*). Due to the overall lower expression of L1rp, ORF2p was barely observable in cells expressing non-recoded L1 (*Figure 1A and B*). ORF1p was mainly present in the cytoplasm but some cells ($\cong$26%) also showed clear nuclear staining (*Figure 1A*, *Figure 1D*, white arrowheads, *Figure 1—figure supplements 1*,*2*, *Videos 1* and *2*). Z-stack movies of cells expressing L1 and stained for recoded and non-recoded ORF1p and ORF2p are also presented (*Videos 1* and *2*). Confocal images recapitulated our observations and confirmed our conclusions (*Figure 1—figure supplement 2*). Interestingly, the cells with nuclear ORF1p fluorescence were usually observed as pairs of cells in close proximity. This observation suggested to us that cells with nuclear ORF1p may have undergone mitotic division immediately prior to fixation and observation. In HeLa cells, this observation was made more evident by the fact that daughter cells displaying nuclear ORF1p were sometimes connected by intercellular bridges often containing filaments of DNA and persisting from incomplete cytokinesis during the previous mitosis (*Figure 1D*, lower panels) (*Steigemann et al., 2009*; *Carlton et al., 2012*). To quantify our initial qualitative observation of closer cell proximity for cells displaying nuclear ORF1p, we employed automated image acquisition of HeLa cells expressing recoded L1 and stained for ORF1p (see 'quantification and statistical analysis' section). We compared the distance of pairs of cells expressing nuclear ORF1p versus the distance of the same

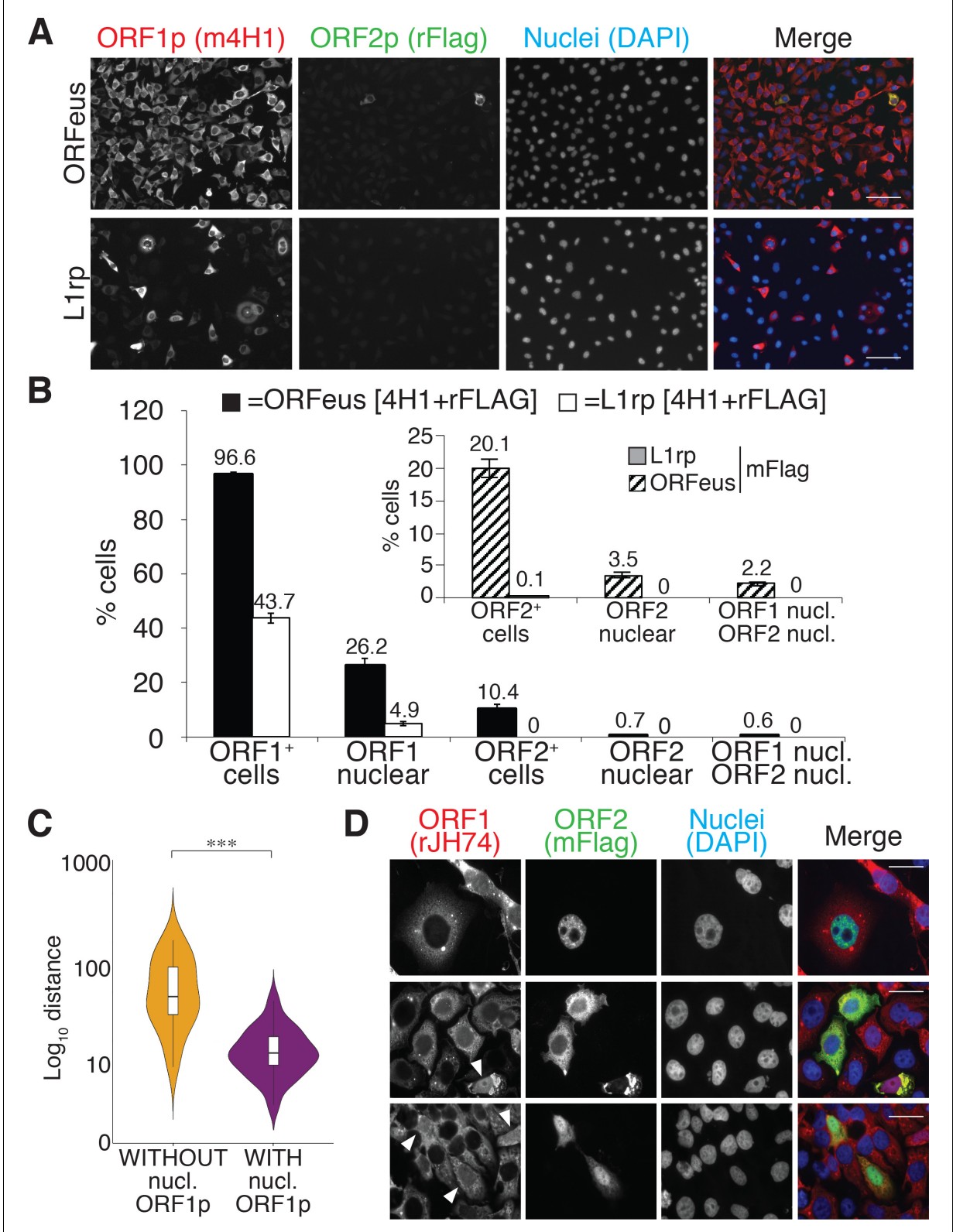

**Figure 1.** L1 protein expression and localization (**A**) Representative pictures of immunostained HeLa M2 cells expressing a recoded L1 (ORFeus) or a non-recoded L1 (L1rp). Antibody target names are reported atop the corresponding pictures and colored according to the colors used in the merged pictures. Scale bar = 100 μm. (**B**) Quantification of ORF1p and ORF2p expression in the cytoplasm and nucleus of HeLa cells expressing the indicated L1 element (error = S.E.M.) of at least 10 20X fields with about 100 cells each. The inset shows quantification of ORF2p stained using an anti-FLAG

*Figure 1 continued on next page*

*Figure 1 continued*

mouse antibody (SIGMA-M2 F1804). (**C**) Proximity analysis of cells with (purple) or without (yellow) nuclear ORF1p is reported. HeLa cells expressing Flag tagged ORFeus for 24 hr, were stained with anti-ORF1p JH74 Ab and DAPI. Pictures were collected using an Arrayscan microscope and analysis was performed as described in the Materials and Methods. ***p<0.001 (p=4.351e$^{-13}$) (**D**) Immunostaining of ORF1p and ORF2p using, respectively, rabbit monoclonal Ab JH74 and mouse monoclonal antibody M2 against FLAG. Scale bar = 20 μm. White arrowheads indicate cells with nuclear ORF1p.

DOI: https://doi.org/10.7554/eLife.30058.003

The following figure supplements are available for figure 1:

**Figure supplement 1.** ORF1p antibody comparison.
DOI: https://doi.org/10.7554/eLife.30058.004

**Figure supplement 2.** Confocal images of LINE-1 ORF1p and ORF2p.
DOI: https://doi.org/10.7554/eLife.30058.005

**Figure supplement 3.** Role of L1 5'UTR in ORF1p and ORF2p localization.
DOI: https://doi.org/10.7554/eLife.30058.006

**Figure supplement 4.** Negative controls for IF staining.
DOI: https://doi.org/10.7554/eLife.30058.007

---

number of cell pairs expressing ORF1p just in the cytoplasm. This 'proximity analysis', described in more details in the methods section, clearly shows that cells with nuclear ORF1p are statistically closer to each other than cells expressing ORF1p exclusively in the cytoplasm (*Figure 1C*).

The cytoplasmic ORF1p localization pattern was often dominated by previously described cytoplasmic foci (*Doucet et al., 2010*; *Goodier et al., 2007*). The formation of these foci is particularly enhanced by L1 overexpression. The nature and role of these cytoplasmic structures is still debated (*Goodier et al., 2007*; *Martin and Branciforte, 1993*; *Guo et al., 2014*) and in our system they may be induced by non-physiological expression levels of ORF1p. We therefore performed the localization experiments reported here using Tet-inducible constructs, and we induced L1 expression at lower concentrations of doxycycline (0.1 μg/ml) compared to the concentrations typically used to induce full induction of retrotransposition or production of RNP intermediates for proteomic studies (1 μg/ml) (*Taylor et al., 2013*). We observed very heterogeneous localization of ORF1p and ORF2p in cells overexpressing L1 upon treatment with 0.1 μg/ml doxycycline for 24 hr. ORF1p is observed to be only cytoplasmic (*Figure 1D* top panel), or both cytoplasmic and nuclear (*Figure 1D*, top panel) but is never exclusively nuclear when immunostaining is performed using 4H1, 4632 and JH74 antibodies (*Figure 1—figure supplements 1,2*). ORF2p also had heterogeneous nuclear/cytoplasmic localization with cells displaying ORF2p only in the cytoplasm (*Figure 1A* and *Figure 1D* middle panel), only in the nucleus (*Figure 1D*, top panel) or in both the cytoplasm and the nucleus (*Figure 1D*, lower panel). Nuclear ORF2p had a punctate pattern (*Figure 1D* top panel) and nuclear ORF1p was usually excluded from nucleoli (*Figure 1D*, arrowheads). Also, a small population of cells (usually less than 0.1% of L1 expressing cells) showed a strong and clear nuclear signal of ORF2p in the absence of nuclear ORF1p (*Figure 1D*, top panel and *Figure 1—figure supplement 2*) consistent with a possible nuclear form of L1 RNPs that lacks ORF1p (see also accompanying paper by *Taylor et al., 2018*). On the contrary, in the cytoplasm, ORF2p always co-localized with ORF1p (*Figure 1A and D*).

We compared several antibodies against ORF1p that confirmed a consistent pattern of nuclear/cytoplasmic localization of ORF1p (*Figure 1—figure supplement 1*). The corresponding secondary-only controls and IF staining of

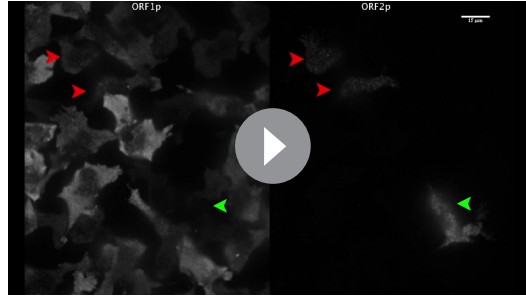

**Video 1.** ORFeus ORF1p/ORF2p Z stack. Z-stacks of recoded L1 (ORFeus) ORF1p (left) and ORF2p (right) expressed in HeLa-M2 treated for 24 hr with doxycycline 0.1 μg/ml. Red arrows points to cells expressing cytoplasmic and nuclear ORF2p and ORF1p; the green arrow points to a cell expressing cytoplasmic and nuclear ORF2p and just cytoplasmic ORF1p.
DOI: https://doi.org/10.7554/eLife.30058.008

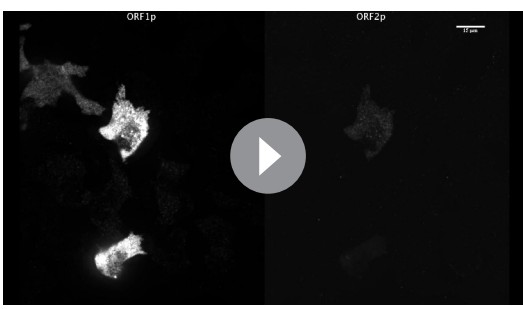

**Video 2.** L1rp ORF1p/ORF2p Z stack. Z-stacks of L1rp ORF1p (left) and ORF2p (right) expressed in HeLa-M2 treated for 24 hr with doxycycline 0.1 µg/ml.
DOI: https://doi.org/10.7554/eLife.30058.009

cells not expressing L1 are presented in *Figure 1—figure supplement 4*. The mouse 4H1 monoclonal antibody (mAb) recognizing the N-terminus of ORF1p and the rabbit JH74 antibody against the C-terminus of ORF1p have an identical pattern. The polyclonal 4632 antibody (pAb) displayed a high nuclear non-specific signal that renders this antibody less sensitive than 4H1 and JH74 (*Figure 1—figure supplements 1,4*). The JH73g rabbit Ab was distinct from the other antibodies and recognized mainly nuclear ORF1p (*Figure 1—figure supplement 1A–B* and *Figure 4—figure supplement 1*). This nuclear form of ORF1p is also recognized by 4H1 and JH74 antibodies but is much more easily observed using JH73g because of its higher affinity for nuclear ORF1p and much lower affin-

ity for cytoplasmic ORF1p. Indeed, quantification of nuclear ORF1p upon staining with JH74 and JH73g revealed a comparable percentage of cells displaying nuclear ORF1p using the two antibodies ($\cong$14–17%). The percentage of cells detected overall as expressing ORF1p using JH73g Ab is much lower than the percentage recognized as ORF1p expressing cells by JH74 Ab (37% versus 93% respectively) because JH73g is able to recognize mainly the nuclear form (*Figure 1—figure supplement 1B*). This unusual staining pattern suggests that the nuclear form of ORF1p may be highly enriched for a conformation specifically recognized by JH73g. Staining of L1rp expressing cells, on the other hand, reveals a lower threshold of sensitivity for the JH73g antibody that is unable to detect the lower amount of non-recoded ORF1p. Interestingly, ORF1p immunoprecipitated with JH73g Ab is impaired in binding to ORF2p protein, consistent with the possibility that most of the nuclear ORF1p species fail to bind ORF2p (*Figure 1—figure supplement 1D*).

## ORF1p enters the nucleus during mitosis

Our immunofluorescence staining of ORF1p and ORF2p suggests that L1 RNPs enter the nucleus during mitosis, when the nuclear membrane breaks down. To better explore this hypothesis, previously put forward by other studies (*Xie et al., 2013*), we exploited well-characterized markers of the cell cycle: geminin, expressed only in S/G2/M phases, and Cdt1, which specifically marks the G1 phase (*Arias and Walter, 2007*). Co-staining of ORF1p with geminin and Cdt1 clearly showed that ORF1p is nuclear in cells expressing Cdt1 (G1 phase), and completely cytoplasmic in cells expressing geminin (*Figure 2A* and quantification in B). Confocal images of Cdt1, geminin and ORF1p staining confirmed our conclusions (*Figure 2—figure supplement 1*). We also verified these results using the FUCCI system (Fluorescent Ubiquitin Cell Cycle Indicator), that exploits Cdt1 and geminin fragments fused to mAG and mKO2 (Monomeric Azami-Green and Monomeric Kusabira-Orange 2 fluorescent proteins) respectively (*Sakaue-Sawano et al., 2008*). Expression of an ORF1p-HaloTag7 in HeLa.S-FUCCI cells treated with the Halo tag ligand JF646, (*Grimm et al., 2015*) shows that ORF1p is nuclear only in cells in G1 phase with orange nuclei resulting from expression of mKO2-tagged Cdt1 fragment (*Figure 2C*). In this setting, no instances of cells with nuclear ORF1p-Halo and green nucleus (cells in S/G2/M) were observed (total number of cells counted = 3309; cells with nuclear ORF1p and green nuclei = 0; cells with nuclear ORF1p and red nuclei = 80). ORF2p-Halo was expressed in cells scattered throughout the cell cycle that displayed either orange (Cdt1 expressing) or green (geminin expressing) nuclei (*Figure 2—figure supplement 2*). This result suggests that ORF2p expression is not cell cycle regulated. Overall, our analyses strongly suggest that ORF1p, most likely together with ORF2p, enters into the nucleus during mitosis and is retained there when the nuclear membrane reforms after cell division.

To gain insight into the nuclear/cytoplasmic distribution of L1 mRNA, we also performed RNA-FISH followed by immunofluorescence staining of ORF1p and ORF2p in cells expressing recoded (ORFeus) or non-recoded L1 (L1rp) (*Figure 3*). As expected, the detected L1 mRNA mostly co-localized with ORF1p staining in both ORFeus and L1rp (*Figure 3A* and additional images in *Figure 3—figure supplement 1A*). Interestingly, and in line with our results on ORF1p localization (*Figure 1C*

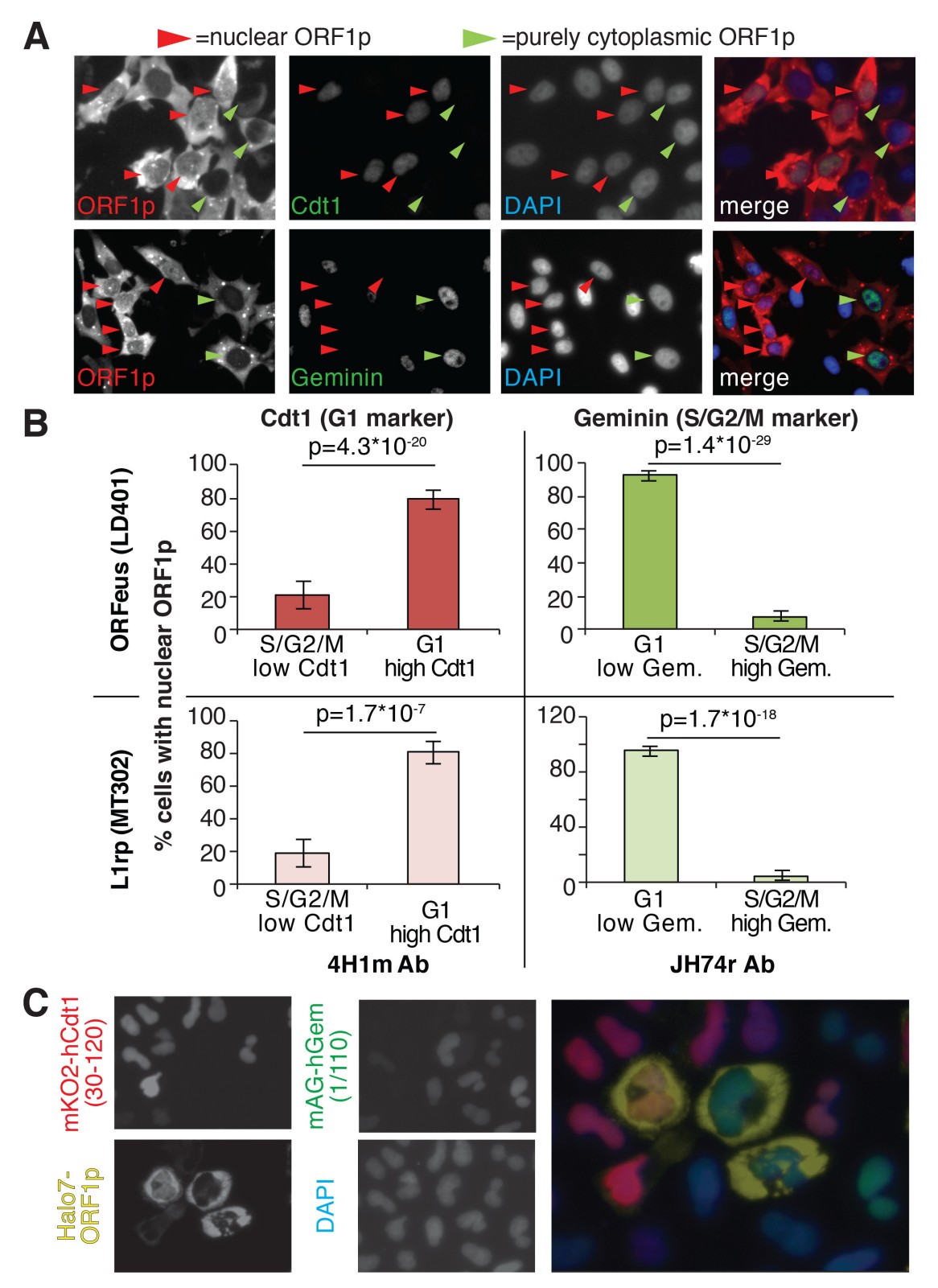

**Figure 2.** ORF1p localizes inside the nucleus immediately after mitosis. (**A**) Representative pictures of co-immunostaining of ORF1p with Cdt1 (marker of G1 phase) or geminin (marker of S/G2/M phase). DNA was stained with DAPI. Red arrowheads = cells with nuclear ORF1p, green arrowheads = cells without nuclear ORF1p. (**B**) Quantification of cells expressing nuclear ORF1p and Cdt1 (red, left top and bottom) or Geminin (green, right top or bottom). HeLa M2 cells expressing ORFeus (plasmid LD401) or L1rp (plasmid MT302) are shown in the two top or bottom panels respectively. The

*Figure 2 continued on next page*

*Figure 2 continued*

antibodies used for staining and quantification are indicated at the bottom of the graphs. (error = S.E.M.). (**C**) Representative pictures of live HeLa.S-FUCCI cell lines expressing rtTA and an inducible recoded L1 with ORF1p C-terminally tagged with an Halotag7 (*Ohana et al., 2009*). JF646 ligand (*Grimm et al., 2015*) was used to visualize ORF1-Halotag7. The proteins visualized are reported to left of images in colors used in the merged picture (right panel).

DOI: https://doi.org/10.7554/eLife.30058.010

The following figure supplements are available for figure 2:

**Figure supplement 1.** Confocal images of ORF1p, Cdt1/geneticin co-staining.

DOI: https://doi.org/10.7554/eLife.30058.011

**Figure supplement 2.** Expression and localization of Halo7-ORF2p in HeLa.S-FUCCI cells.

DOI: https://doi.org/10.7554/eLife.30058.012

and *Figure 2*), nuclear signal of L1 mRNA was particularly bright in cells exiting mitosis identified by cytoplasmic bridges still connecting the two daughter cells (*Figure 3—figure supplement 1B*, *Figure 3—figure supplement 2* and *Videos 3* and *4*). To verify that high nuclear L1 mRNA was detected in post-mitotic cells in G1 phase, we performed RNA-FISH of ORFeus followed by Cdt1 and Geminin co-staining (*Figure 3B–C* and *Figure 3—figure supplement 2*). As shown for ORF1p and in line with our hypothesis, nuclear L1 mRNA co-localized with cells with low Geminin and higher Cdt1 staining. These results confirmed our ORF1p localization results, and suggest that L1 RNPs formed by ORF1p, L1 mRNA and, most likely, ORF2p enter the nucleus during mitosis and remain 'trapped' in the nucleus upon nuclear membrane reformation in G1 phase.

A direct consequence of these conclusions would be that prolonged expression of L1 in dividing cells should eventually lead to a population with all cells displaying nuclear ORF1p because a longer time of L1 induction will allow all the cells to undergo mitosis while expressing ORF1p. To test this hypothesis, we quantified the percentage of cells displaying nuclear ORF1p after 24 and 48 hr of L1 expression, considering that HeLa cell doubling time is about 24 hr. Automated picture collection (Arrayscan HCS, Cellomics) and software based nuclear/cytoplasmic analysis (HCS studio cell analysis software) was implemented as for proximity analysis. We set very stringent negative fluorescence thresholds (limit in *Figure 4A*) determined from cells not treated with doxycycline and therefore not expressing L1, as described in the methods section. These stringent parameters were necessary to avoid interference of the strong ORF1p cytoplasmic signal with the measurement of nuclear ORF1p signal. The analysis of ORF1p nuclear and cytoplasmic distribution surprisingly showed that the percentage of cells with nuclear ORF1p does not increase but actually decreases after 48 hr of L1 induction compared to the 24 hr time point (*Figure 4A*). The decrease in nuclear ORF1p after 48 hr induction is probably due to the decreased growth rate of a more confluent cell population. The absence of an increase of cells with nuclear ORF1p with increased time of L1 induction, suggests that, after entering the nucleus in M phase, nuclear ORF1p is either degraded and/or exported from the nucleus during or after G1 phase.

## ORF1p nuclear localization is increased upon leptomycin treatment

To better explore potential cytoplasmic/nuclear shuttling of ORF1p and ORF2p we took advantage of a known inhibitor of exportin 1 (XPO1/CRM1), leptomycin b. We treated HeLa cells expressing LINE-1 with leptomycin for 18 hr. Two different concentrations of leptomycin were used and several antibodies (Abs) were utilized to detect ORF1p in immunofluorescence assays (*Figure 4B–E*). At both leptomycin concentrations, and using any of the Abs recognizing ORF1p we observed an increased number of cells with nuclear ORF1p after leptomycin treatment, suggesting that at least a subset of ORF1p is exported from the nucleus in a CRM1-dependent manner (*Figure 4E*). As control, a known CRM1 regulated protein (MEK-1) (*Dave et al., 2014*) tagged with GFP was used to show nuclear retention upon leptomycin treatment (*Figure 4—figure supplement 2*).

## LINE-1 retrotransposition peaks during S phase

Our results suggest that ORF1 protein, in a ribonucleoprotein complex with L1 mRNA (and presumably ORF2p), is able to enter the nucleus during mitosis and it accumulates in the nucleus in early G1 phase of the cell cycle. Following early G1, ORF1p is then exported to the cytoplasm through a CRM1 dependent mechanism. We therefore asked whether L1 retrotransposition occurred in a cell

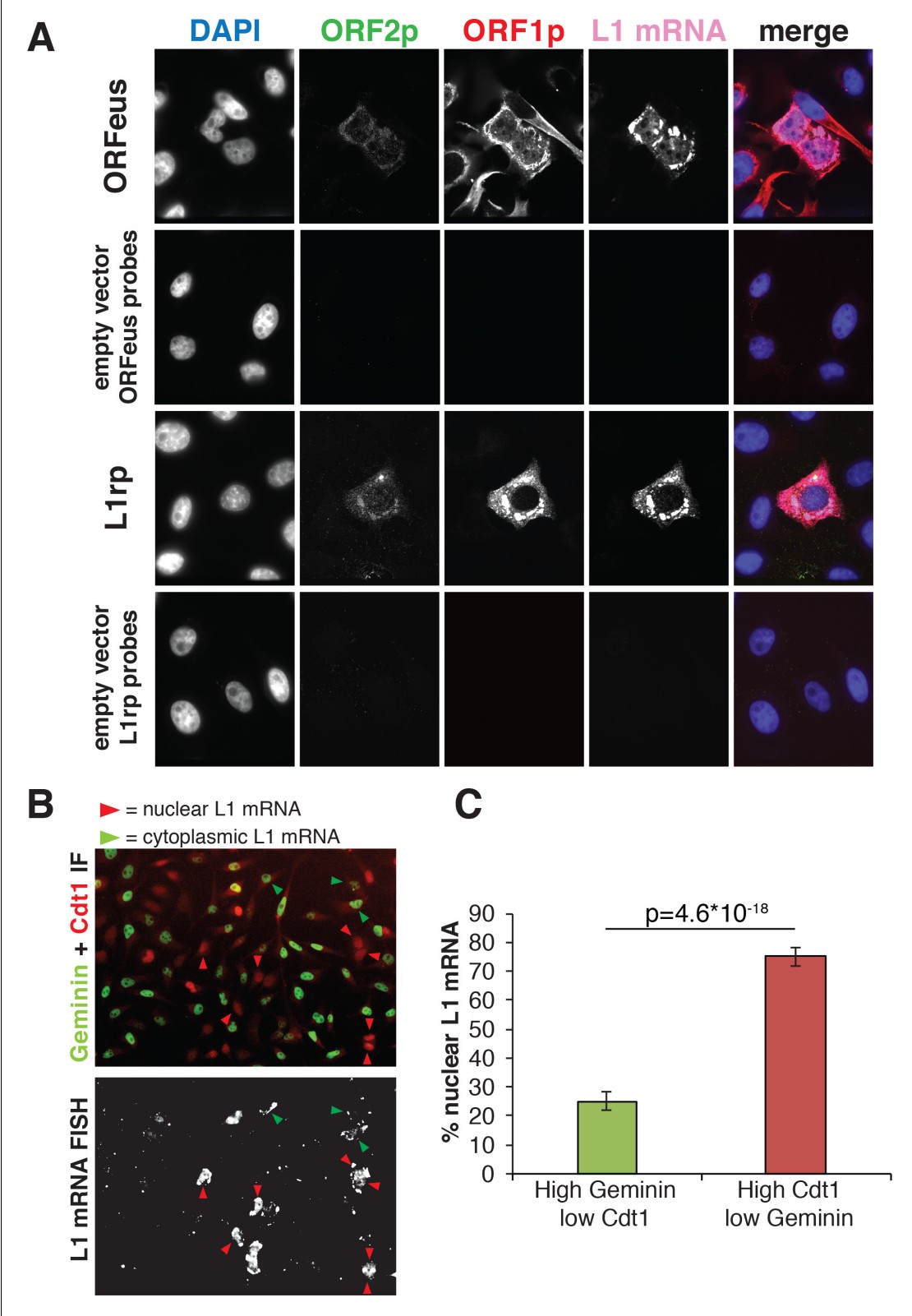

**Figure 3.** Analysis of L1mRNA nuclear/cytoplasmic localization. (**A**) HeLa-M2 cells expressing ORFeus (row 1), L1rp (row3) or not expressing L1 (row 2 and 4, controls) were stained for L1mRNA using cy5 conjugated probes against ORFeus or L1rp. IF of ORF1p using rabbit JH74 Ab and ORF2p using mouse FLAG-M2 Ab was performed right after RNA-FISH labelling. Z-stack images were collected using a spinning disk confocal microscope. The max intensity projection images of each fluorescence channel are shown in grayscale together with the merge image in pseudocolors. (**B**) L1 mRNA was

*Figure 3 continued on next page*

*Figure 3 continued*

stained using RNA-FISH together with IF staining of Cdt1 and geminin, makers of G1 and S/G2/M respectively. HeLa-M2 cells expressing recoded L1 (pCEP-puro plasmid LD401) were used. A merged image of Cdt1 (red) and geminin (green) IF is shown in the top image while L1 RNA-FISH is presented in grey-scale in the bottom image. Red arrowheads indicate cells that show strong nuclear L1 mRNA signal and are in G1 phase (low Geminin, high Cdt1). Green arrowheads indicate cells that show only cytoplasmic L1 mRNA signal and that are in S/G2/M phase (high Geminin, low Cdt1). Magnified pictures of some representative cells pointed by red and green arrows are presented in *Figure 3—figure supplement 2*. (C) Quantification of the distribution of nuclear L1 mRNA among G1 or S/G2/M cells. (error bar = S.D., n = 4). Two tail T-test p value is reported.

DOI: https://doi.org/10.7554/eLife.30058.013

The following figure supplements are available for figure 3:

**Figure supplement 1.** Additional confocal images of L1 mRNA detected by RNA-FISH.

DOI: https://doi.org/10.7554/eLife.30058.014

**Figure supplement 2.** Details of *Figure 3B*.

DOI: https://doi.org/10.7554/eLife.30058.015

cycle-dependent manner and more specifically during M phase or G1 phase, when we observed ORF1p in the nucleus and when chromatin is accessible to L1 RNPs. To answer this question we performed retrotransposition assays using a previously described ORFeus-GFP-AI reporter (*Taylor et al., 2013*; *An et al., 2011*). HeLa cells expressing the retrotransposition reporter were treated for increasing times with nocodazole (*Figure 5A*), a cell cycle inhibitor that blocks cells in M phase interfering with microtubule assembly (*Ma and Poon, 2017*; *Rosner et al., 2013*). Treatments were performed for no longer than 21 hr, a time sufficient to allow cells passage through just one cell cycle. Increased time of nocodazole treatment, and therefore longer time in M phase, fails to increase the percentage of M phase green cells (*Figure 5A–B*), suggesting that L1 retrotransposition does not occur during M phase. Longer times of nocodazole treatment (21 hr) increased cell death, detected by an increase of propidium iodide-positive cells, and a consequent decrease in retrotransposition (*Figure 5B*, dotted line). Similar experiments were also performed using thymidine and mimosine treatments to interrogate possible biases of L1 retrotransposition toward G1 phase (*Ambrozy, 1971*; *Lalande, 1990*). The effects on cell cycle progression of increased times of 4 mM thymidine and 1 mM mimosine treatments are reported in *Figure 5—figure supplement 1*. Treatment with excess thymidine inhibits DNA synthesis blocking cells in late G1. As with nocodazole treatments, cells treated with thymidine, showed no increase in GFP positive cells compared to untreated cells, suggesting that L1 does not preferentially retrotranspose in late G1 phase (*Figure 5C*). Mimosine treatments, also did not increase retrotransposition but actually decreased L1 hopping (*Figure 5D*). Mimosine is a non-protein amino acid that potently inhibits cell cycle. Despite mimosine's well-established role in blocking cells in late G1/early S phase, the molecular mechanisms affected by mimosine to induce cell cycle arrest are still debated. Mimosine was shown to affect both DNA synthesis initiation and elongation and to induce depletion of deoxynucleotides through chelation of iron and consequent inhibition of ribonucleotide reductases (RNR) and serine hydroxymethyltransferase (SHMT). Mimosine was also shown to inhibit viral replication consistent with a general metabolic mechanism (*Nguyen and Tawata, 2016*; *Kalejta and Hamlin, 1997*; *Dai et al., 1994*; *Park et al., 2012*). Our data showing a decreased retrotransposition in cells treated with mimosine, suggest that mimosine inhibits L1 retrotransposition not only arresting the cell cycle but probably through additional (metabolic?) mechanisms (*Figure 5D*).

We then expanded our analysis, measuring retrotransposition during a single cell cycle in a population of HeLa cells synchronized by nocodazole treatment, subsequent 'mitotic shake off' and released into the cell cycle in the absence of nocodazole (*Figure 6A*). Measurements of the percent of cells that underwent retrotransposition were performed every three hours starting after release from nocodazole synchronization.

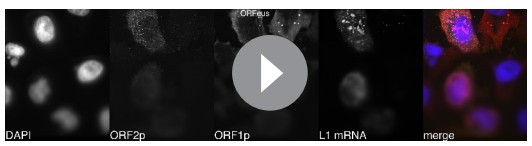

**Video 3.** ORFeus ORF1p/ORF2p/L1 mRNA Z stack. Z-stacks of chromatin (DAPI), ORF2p (FLAG), ORF1p (JH74) and L1 ORFeus mRNA (cy5-probes) in HeLa-M2 cells expressing ORFeus (LD401) and treated for 24 hr with doxycycline 0.1 μg/ml. The max intensity projection images of these cells is presented in *Figure 3A* row 1.

DOI: https://doi.org/10.7554/eLife.30058.016

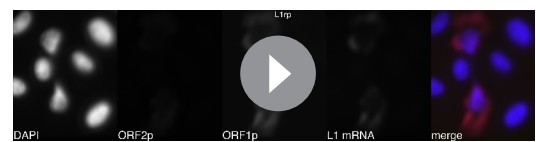

**Video 4.** L1rp ORF1p/ORF2p/L1mRNA Z stack. Z-stacks of chromatin (DAPI), ORF2p (FLAG), ORF1p (JH74) and L1rp mRNA (cy5-probes) in HeLa-M2 cells expressing L1rp (MT302) and treated for 24 hr with doxycycline 0.1 μg/ml. The max intensity projection image of these cells is presented in *Figure 3A* row 4.
DOI: https://doi.org/10.7554/eLife.30058.017

The cell cycle stage of the cells at each time point was determined by propidium iodide staining (*Figure 6—figure supplement 1*). A linear increase of retrotransposition should be observed if retrotransposition is unbiased towards specific cell cycle stages, while a non-linear increase represents a specific stage at which retrotransposition is enhanced. Calculation of the slope of the increase of GFP+ cells should therefore produce a clear peak at the time during which most retrotransposition occurs (*Figure 6D*). This approach allowed us to identify a peak of retrotransposition in the S phase (*Figure 6E* top and bottom left panels). Control non-synchronized cells, as expected, showed a linear increase in retrotransposition and no clear peaks were identified (*Figure 6F* top, bottom right panels).

To evaluate whether the cell cycle controls retrotransposition using a method independent of cell synchronization, we developed a fluorescent-AI reporter that introduces a temporal component to canonical retrotransposition reporters. To this end, we utilized the previously characterized monomeric fluorescent timers (FT) (*Subach et al., 2009*). These derivatives of mCherry change their fluorescence emission from blue to red over 2 to 3 hr (fast-FT). We introduced an antisense intron within the coding region of the 'fast-FT' and inserted this cassette in the 3'UTR of a recoded L1 (*Figure 7A*). Transfection of the L1-fastFT-AI construct into HeLa cells allowed us to identify cells that underwent retrotransposition within a ~ 3 hr period preceding the analysis, as reported by previous work (*Subach et al., 2009*). Our quantification also supports the previously reported timing of FT maturation (*Subach et al., 2009*) with an average conversion time from blue to red of 2.35 ± 0.52 hr (*Figure 7A* and *Figure 7—figure supplement 1B*). Immediately after L1-fastFT-AI retrotransposition, the fast FT is expressed and the cells emit blue fluorescence. Upon translation the blue proteins begin turning red in less than 3 hr. To roughly quantify the time needed for the visualization of a fluorescent protein after induction of transcription, we measured GFP expression upon doxycycline induction. Quantification of cells expressing GFP under control of a Tet CMV-inducible promoter revealed that 50% of the cells expressed visible GFP at 2.71 ± 0.46 hr and 90% of the cells expressed visible GFP within 8.01 ± 0.63 hr of doxycycline treatment (*Figure 7A* and *Figure 7—figure supplement 1A*). This quantification, even when performed in an over-expression setting, suggests that transcription from a strong promoter, translation and accumulation of a fluorescent protein can be fast enough for the detection of retrotransposition events within approximately 3 hr from the event itself.

In cells expressing L1 FT-AI, upon retrotransposition of L1, the FT gene is transcribed and expressed in <3 hr in at least 50% of the expressing cells. After 2.35 ± 0.52 hr from translation, the blue proteins mature into red emitting proteins. The cells are now marked by a blue population of proteins continuously transcribed by the constitutive CMV promoter and a red population of aged proteins matured from the blue form (*Figure 7A*). Analysis of the cell cycle stage of FACS sorted 'blue only' cells (cells that underwent retrotransposition within about 3 hr of the analysis, also considering the time needed for transcription of the marker) compared to fluorescence negative cells FACS sorted from the same population (cells that did not undergo retrotransposition before analysis) revealed a strong enrichment in S phase cells, a partial enrichment in cells in the G2/M phase and a strong de-enrichment of cells in G1 phase (*Figure 7B* and *Figure 7—figure supplement 2A–B*). These results confirmed the strong bias of retrotransposition towards S phase that we measured using nocodazole synchronization (*Figure 6*). However, the over-representation of G2/M cells that underwent retrotransposition pushed us to design experiments to better dissect the population of cells comprising the peak of blue-only cells presented in *Figure 7B*. To better dissect the cell cycle stage of cells that underwent retrotransposition, we implemented a second approach that does not involve cell sorting but simply allows direct analysis of the cell cycle in cells expressing the FT-AI reporter (*Figure 7C* and *Figure 7—figure supplement 2C–D*). After 24 hr of doxycycline treatment, cells expressing the L1-fastFT-AI reporter were directly stained with SYTO61 DNA labeling dye and analyzed. The main population of blue negative cells that did not undergo retrotransposition,

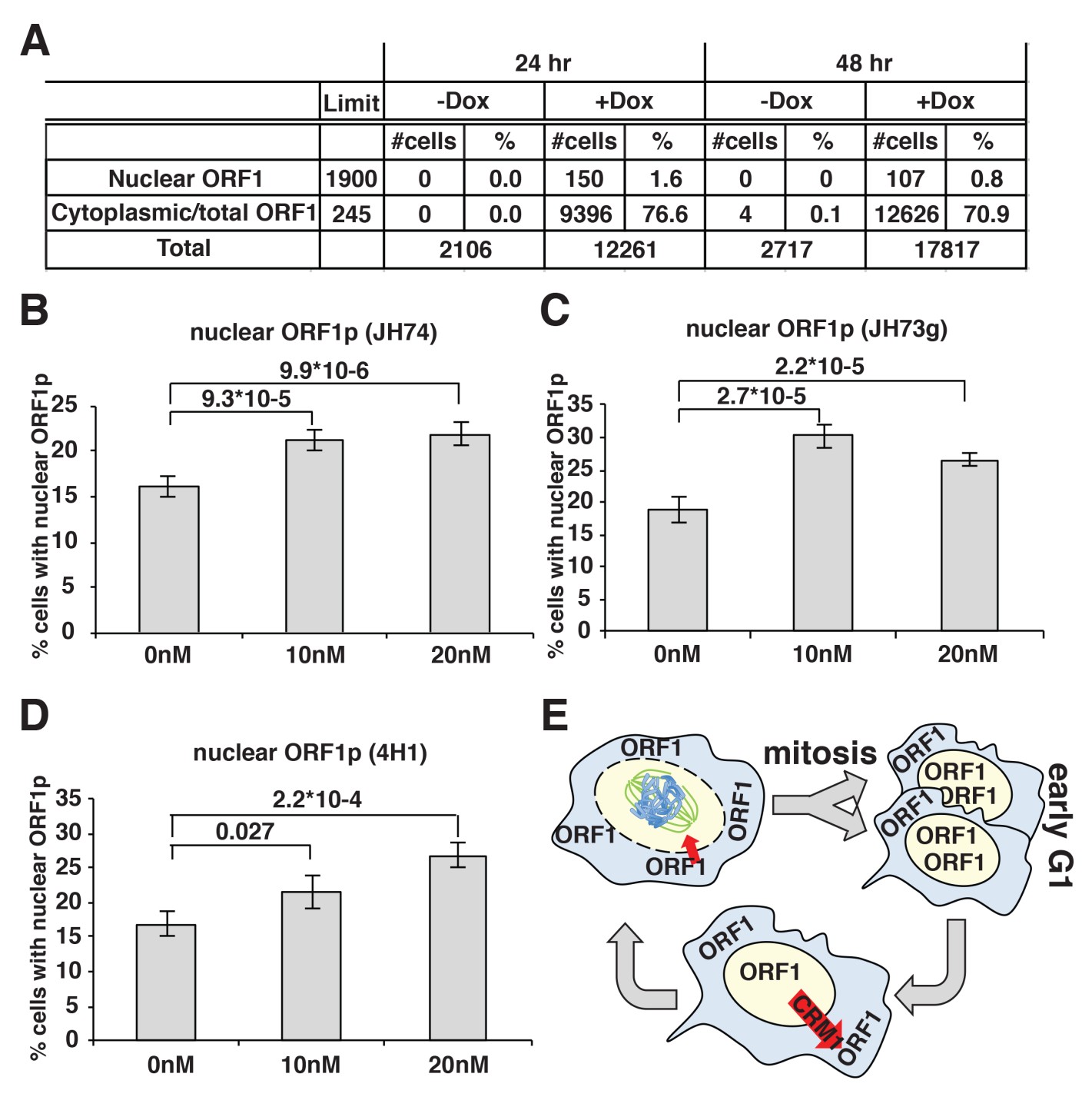

**Figure 4.** ORF1p nuclear localization upon leptomycin treatment. (**A**) HeLa M2 cells expressing a recoded L1 (ORFeus) with Flag tagged ORF2p were treated for 24 or 48 hr with or without 0.1 µg/ml doxycycline on chamber slides. After treatment, cells were fixed in formalin and stained with JH74 primary antibody, Alexa 647 labeled secondary antibody and DAPI. Slides were scanned with Arrayscan and analyzed using Image Studio HCS software. The number of cells with cytoplasmic ORF1p (also considered as the total number of ORF1p expressing cells) and nuclear ORF1p are reported as well as total amount of cells calculated from DAPI staining. (**B–D**) quantification of cells expressing nuclear ORF1p in HeLa M2 cells treated with or without leptomycin 10 or 20 nM for 15 hr. After treatment cells were fixed and ORF1p stained using the indicated antibodies. Two tailed T-test p-values are reported for significant differences (error = S.E.M.) (**E**) Schematic of ORF1p nuclear/cytoplasmic dynamics during the cell cycle.
DOI: https://doi.org/10.7554/eLife.30058.018

The following figure supplements are available for figure 4:

*Figure 4 continued on next page*

*Figure 4 continued*

**Figure supplement 1.** Comparison between JH73 and JH73g antibodies.
DOI: https://doi.org/10.7554/eLife.30058.019
**Figure supplement 2.** Leptomycin treatments of MEK1 expressing cells.
DOI: https://doi.org/10.7554/eLife.30058.020

showed cells distributed throughout the cell cycle (G1 = 49.5%, S = 34.2%, G2/M = 15.2%) (*Figure 7C*, black line and *Figure 7—figure supplement 2C–D*). Using this analysis, we were able to divide the population of blue positive cells (blue$^+$) in two subpopulations: cells with relatively higher red fluorescence (*Figure 7C*, purple profile and *Figure 7—figure supplement 2C–D*), and cells with lower/undetectable red fluorescence (*Figure 7C*, blue profile and Fig. *Figure 7—figure supplement 2C–D*). The former group of cells underwent retrotransposition in a time closer to the time of analysis compared to the blue$^+$ cells with higher red fluorescence in which few FT molecules had time to mature into the red form. Consistent with the previous experiments, the blue$^+$ cells with lower red fluorescence (blue peak) are mainly in S phase (G1 = 9.38%, S = 78.1%, G2/M = 12.5%). Blue$^+$ cells with higher red fluorescence (purple peak), which had more time to proceed through the cell cycle after retrotransposition and before analysis (from S to G2/M), were mainly in G2/M phase (G1 = 0%, S = 10.9%, G2/M = 89.1%). This result clearly shows that the wide peak of 'blue only' sorted cells that spread across S and G2/M phases (*Figure 7B*) actually consists of two subpopulations/peaks: a population of cells in S phase that underwent retrotransposition a short time before analysis and a second population of cells in G2/M phase that underwent retrotransposition earlier relative to analysis.

These observations, together with the data presented in *Figures 5* and *6*, collectively indicate that L1 retrotransposition has a strong cell cycle bias and preferentially occurs during the S phase.

## ORF2p binds chromatin and localizes at replication forks with PCNA during S phase

To gain biochemical insight into the timing of L1 retrotransposition we investigated the timing with which ORF2p was recruited onto chromatin, a necessary step for retrotransposition. We isolated nuclear soluble and chromatin bound proteins from cells synchronized and released into the cell cycle, as in *Figure 6C*. Immunoblot analysis showed no differences in the amount of histone H3 and ORF1p present on chromatin and, as expected, the analysis revealed chromatin recruitment that peaked in S phase for PCNA (*Strzalka and Ziemienowicz, 2011*) and Upf1 (*Azzalin and Lingner, 2006*). Supporting our previous results, ORF2p was recruited on chromatin in S phase in a similar manner to Upf1 and PCNA (*Figure 8A–B*). It is worth noting that, despite our observation that ORF1p is less nuclear in late G1 (*Figure 4*), we did not observe changes in nuclear and cytoplasmic ORF1p during cell cycle progression (*Figure 8A*, right panel). This is probably due to the contamination of cytoplasmic stress granules (highly enriched in ORF1p) in nuclear fractions as shown by detection of G3BP1, a marker of stress granules (*Figure 8—figure supplement 1*).

We previously showed that ORF2p binds PCNA through a PIP domain in the ORF2 protein, sandwiched between the EN and RT domains, and that the PCNA-ORF2p interaction is necessary for retrotransposition in HeLa and HEK293 cells (*Taylor et al., 2013*). The PCNA-ORF2p complex is mainly chromatin bound (*Figure 8—figure supplement 1*), supporting the idea that PCNA binds ORF2p during retrotransposition. We therefore followed up on these previous findings investigating the interactome of the ORF2p-PCNA complex specifically. To this end, we engineered a V5-tag at the N-terminus of PCNA in HCT116 cells stably expressing a doxycycline inducible ORFeus. HCT116 cells were chosen because of their near-diploid number of chromosomes compared to HeLa cells, characterized by unstable karyotype. We performed sequential immunoprecipitation of ORF2p followed by V5/PCNA IP (*Figure 9A*), and we analyzed the interacting partners of the ORF2p-PCNA complex by mass spectrometry. Among the identified ORF2p/PCNA interactors (279 from the first experiment and 158 from the second experiment) we identified several MCM proteins (MCM3, MCM5 and MCM6) as well as TOP1 (DNA topoisomerase 1), PARP1 (Poly [ADP-ribose] polymerase 1) and RPA1 (Replication Protein A1) (*Figure 9B*). These proteins are known to be co-recruited with PCNA on the origins of DNA replication before S phase (MCM proteins) and during S phase on the replication fork (MCM, PCNA, TOP1, RPA1 and PARP1 proteins) (*Czubaty et al., 2005*;

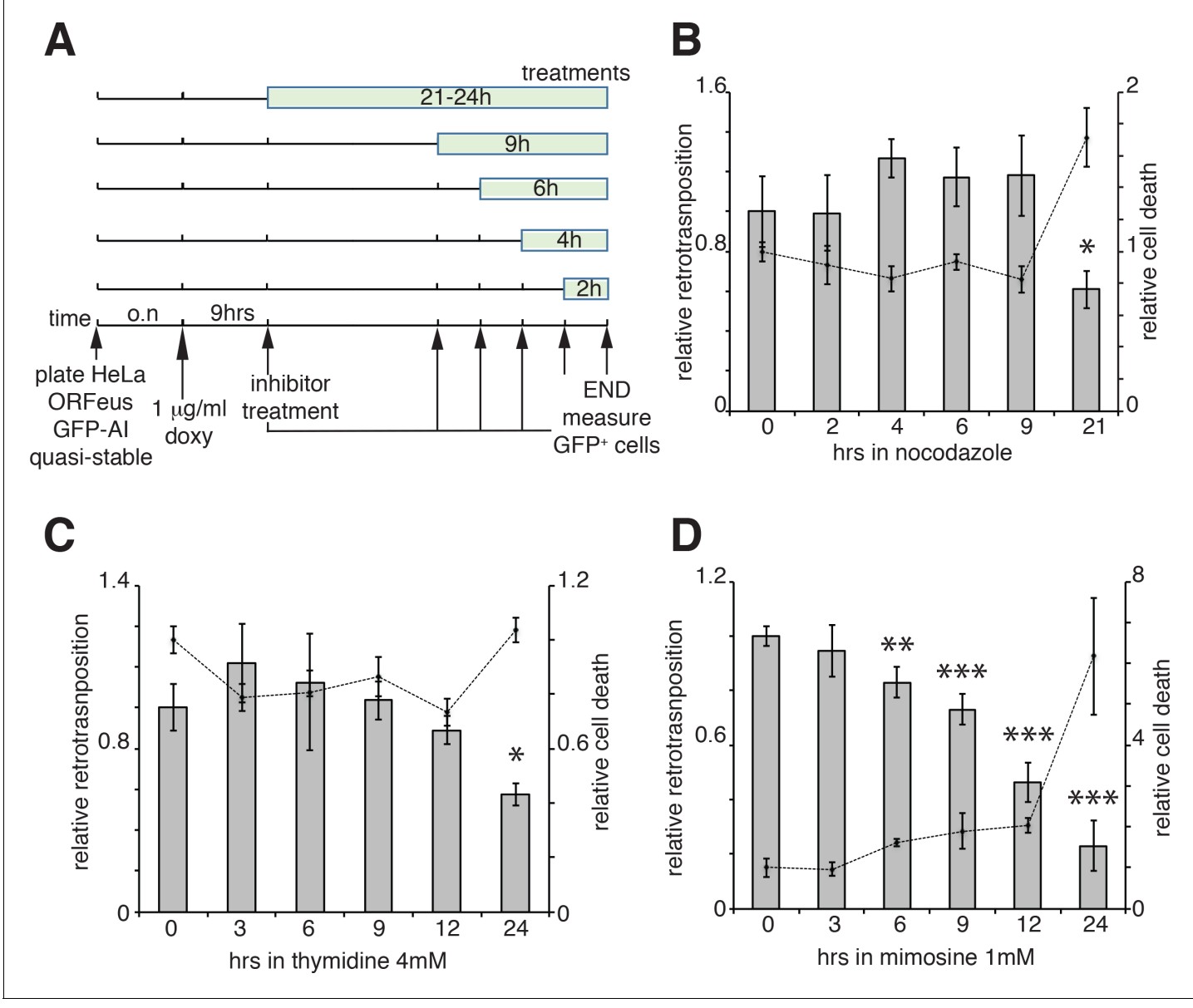

**Figure 5.** Retrotransposition in cells blocked in M and G1 phases. (**A**) Scheme of the experimental timeline followed for experiments presented in b-d). HeLa M2 cells stably expressing episomal ORFeus-GFP-AI reporter (EA79 plasmid) were plated in 10 cm tissue culture treated plates (3 cm wells in c and d). After overnight incubation, expression of L1 was induced with 1 μg/ml doxycycline. After 9 hr induction, nocodazole, thymidine or mimosine were directly added to the media at various times in different plates to a final concentration of 60 ng/ml nocodazole, 4 mM thymidine and 1 mM mimosine. After 30 hr from the beginning of doxycycline treatment M phase cells were collected by shake-off (all cells were collected by trypsinization for c and d) and the percentage of GFP+ cells measured using a flow cytometer as described in the method section. (**B–D**) Histogram boxes (left Y axes) represent relative % of GFP+ cells measured by cytofluorometry after treatments with nocodazole (**B**), 4 mM thymidine (**C**) or 1 mM mimosine (**D**) as described in A). The dots connected by a dotted line (right Y axes) represent the relative % of PI+ cells (dead cells). Retrotransposition and PI percentages of cells not treated with nocodazole, thymidine or mimosine (time 0) were set as 1. (error = S.D., *p<0.05; **p<0.01; ***p<0.001).
DOI: https://doi.org/10.7554/eLife.30058.021

The following figure supplement is available for figure 5:

**Figure supplement 1.** Cell cycle profiles of cells treated with thymidine and mimosine.
DOI: https://doi.org/10.7554/eLife.30058.022

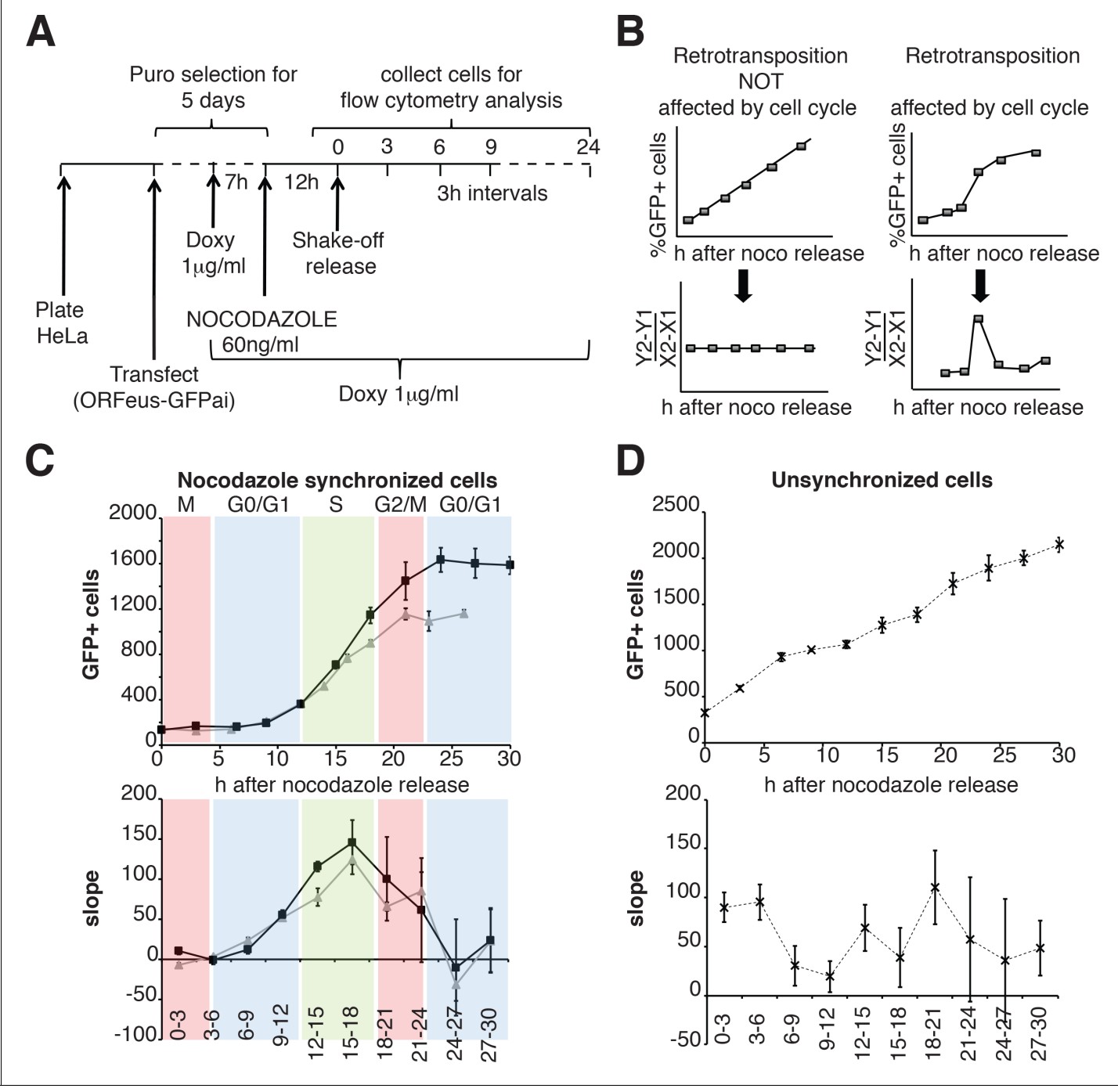

**Figure 6.** Retrotransposition during the cell cycle. (**A**) Scheme of the experimental timeline followed for experiments presented in E-F). HeLa M2 cells were plated and transfected in six well plates. 24 hr after transfection 1 μg/ml puromycin was added to the medium for 5 days. During puromycin selection cells were split into 10 cm plates to avoid contact inhibition. During the 5th day of puromycin selection, $3 \times 10^6$ cells were freshly replated in 10 cm plates and after 3 hr, doxycycline was added to the medium to a final concentration of 1 μg/ml. After 7 hr, nocodazole was added to a final concentration of 60 ng/ml. After 12 hr, medium was discarded and mitotic cells were collected by mechanical shake off. Cells were washed and $0.4 \times 10^6$ mitotic cells were replated in each of 3 cm wells. At the indicated time points, the percentage of GFP+ cells was measured using a flow cytometer. (**B**) prediction of retrotransposition measurements if cell cycle affects (left panels) or does not affect (right panels) retrotransposition. (**C–D**) Retrotransposition analysis of nocodazole synchronized (**C**) or not synchronized (**D**) cells. The % of GFP+ cells in a population of 10000 cells is reported in the indicated time points. Bottom panels show slope changes from the corresponding measurements on the top panels. The shaded colored boxes indicate specific cell cycle stages extrapolated from propidium iodide (PI) measurements reported in *Figure 6—figure supplement 1*. The two lines (gray and black) represent two experiments using independent transfections of the retrotransposition reporter. (error = S.D., n = 4).

*Figure 6 continued on next page*

*Figure 6 continued*

DOI: https://doi.org/10.7554/eLife.30058.023

The following figure supplement is available for figure 6:

**Figure supplement 1.** PI analysis of the cell cycle.

DOI: https://doi.org/10.7554/eLife.30058.024

*Remus et al., 2009*; *Ying et al., 2016*). Co-immunoprecipitation of Flag/ORF2 or ORF1 proteins from HEK293 cells expressing ORFeus, recapitulated the interaction of ORF2p with MCM6 and PCNA (*Figure 9B*). As expected, immunoprecipitation of ORF2p pulled down a fraction of ORF1p, but also MCM6 and PCNA proteins. Interestingly, as expected from our previous observations revealing that ORF1p is not necessarily in the complex(es) with chromatin bound nuclear L1 RNPs, immunoprecipitation of ORF1p pulled down only a small amount of ORF2p, and also a smaller amount of MCM6 and PCNA proteins. These observations suggest that the nuclear L1 complex contains ORF2p, PCNA and components of the replication fork such as MCM6, and is depleted of ORF1 proteins.

To verify that the ORF2p-PCNA-MCM complex identified here also contained L1 mRNA, component of the L1 RNPs essential for retrotransposition, we performed IP experiments followed by RT-qPCR (IP-RT-qPCR) for L1. As expected, IP of ORF2p pulled down L1mRNA (*Figure 9D*, top panel) as well as direct IP of PCNA also showed interaction of this protein with the L1 mRNA (*Figure 9D*, top panel) compared to control IPs performed using normal mouse IgG antibodies. In line with our hypothesis that the ORF2p-PCNA complex is potentially retrotransposing, sequential IP of ORF2p followed by PCNA IP also pulled down L1 mRNA (*Figure 9D*, lower panel) compared to control IgG IP. Control qPCR of samples not treated with reverse transcriptase (-RT) displayed no or extremely low amplification (data not shown).

Finally, we performed immunofluorescence staining of ORF2p and PCNA in HeLa cells synchronized in S phase by double thymidine synchronization. A subset of nuclear ORF2p puncta overlapped with PCNA foci, marking potential regions of active DNA replication (*Figure 9D*).

Collectively, our biochemical and proteomic work (*Figure 9*) support the hypothesis that ORF2p binds PCNA on sites of DNA replication during S-phase (model in *Figure 10*), and this fraction is most likely engaging in retrotransposition as demonstrated by our functional assays (*Figure 7*).

## Discussion

Despite the increasingly appreciated relevance of L1 retrotransposon to normal cellular physiology and disease etiology, many of the steps of L1 retrotransposon lifecycle in human cells are largely unknown. This lack of insight about L1 retrotransposons in human cells is unsurprising considering that many technical challenges hinder studies of this highly repetitive but poorly expressed element, which is effectively repressed by host somatic cells (*Goodier, 2016*) and therefore overexpression approaches can reveal otherwise hidden pathways.

It makes sense that nuclear localization of a L1 RNP particle comprising at least ORF2p, with EN and RT activity, bound to L1 mRNA, is essential for L1 retrotransposition to gain access to its target, genomic DNA. Despite this obvious observation, the nature of nuclear L1 RNPs and the process by which L1 gains entry into the nucleus are unknown. No functional nuclear localization signal (NLS) has been identified in the two L1 proteins ORF1p and ORF2p suggesting that their import into the nucleus is either mediated by interacting partners or by cellular processes such as the cell cycle and progression through mitosis during which the nuclear membrane breaks down, allowing the possible entrance of L1 RNPs into the nucleus. The former hypothesis is supported by several studies that show an essential role of cell division on retrotransposition and retrotransposition rate in tissue culture cells (*Xie et al., 2013*; *Shi et al., 2007*). On the other hand, other work showed the possibility of L1 retrotransposition in differentiated and non-dividing cells such as human neurons and glioma cells, albeit at substantially lower rates (*Macia et al., 2017*; *Kubo et al., 2006*). These seeming incongruities may be explained with a possible major mechanism of entry into the nucleus during mitosis and a less frequent mode of nuclear localization for L1 RNPs that is independent of the cell cycle and specific for some cellular state or cell type. Other possible explanations for the

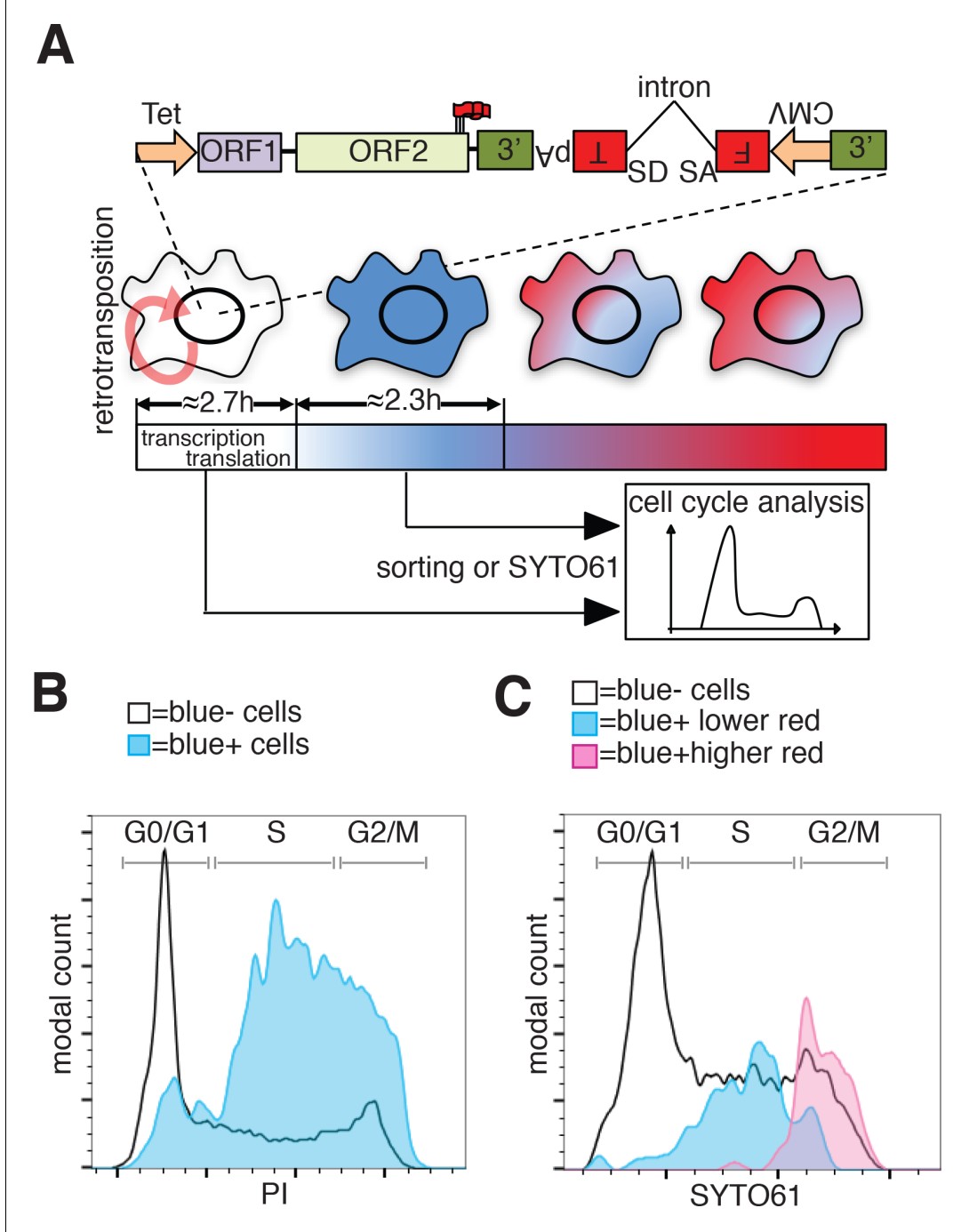

**Figure 7.** Analysis of L1 retrotransposition during the cell cycle using FT-AI reporter. (**A**) Schematic of the fluorescent-timer-AI reporter (FT-AI) and of the experimental design. HeLa M2 cells were transfected with a FT-AI reporter. After induction of L1 expression (24 hr, 1 μg/ml doxycycline), cells undergoing retrotransposition are blue and can be sorted by FACS or directly stained with SYTO61 for cell cycle analysis. Within about 2.7 hr of doxycycline treatment 50% of the cells will start to express the FT reporter as estimated in *Figure 7—figure supplement 1A*. After about 2.35 hr (*Figure 7—figure supplement 1B*) from expression of the blue FT (retrotransposition event), cells start to become red. Double negative (blue-/red-) cells were also collected by sorting and analyzed as control. The sorted cells are then stained with PI and their cell cycle stage determined. (tet = Tetracycline inducible promoter, 3'=3'UTR, pA = polyA signal, FT = fluorescent timer, SD = splice donor, SA = splice acceptor, CMV = cytomegalovirus constitutive promoter, red round arrow = L1 'jumping'). (**B**) Histogram of cell cycle distribution of sorted and PI stained blue⁻red⁻ cells (black line) and blue⁺red⁻ cells (blue

*Figure 7 continued on next page*

*Figure 7 continued*

histogram). The percentage of cells in each cell cycle stage after sorting and PI staining are reported in *Figure 7—figure supplement 2B*. (C) Cell cycle analysis using SYTO61 dye of blue⁻ cells that did not undergo retrotransposition (black line), of blue⁺ cells expressing lower red signal (blue histogram) or blue⁺ cells expressing higher red signal (purple histogram). The complete profiles of analysis for the reported cells are presented in *Figure 7—figure supplement 2C–D*.

DOI: https://doi.org/10.7554/eLife.30058.025

The following figure supplements are available for figure 7:

**Figure supplement 1.** Dynamics of GFP expression, fast FT maturation and PI analysis.
DOI: https://doi.org/10.7554/eLife.30058.026
**Figure supplement 2.** Cell cycle analysis of FT-AI.
DOI: https://doi.org/10.7554/eLife.30058.027

discrepancies between our conclusions and works showing retrotransposition in non-dividing cells, are potential cell cycle artifacts caused by the adenovirus vectors or the low rate of proliferation of the cells used.

Through imaging, genetic and biochemical approaches, we show that L1 nuclear import as well as L1 retrotransposition has a strong cell cycle bias (*Figure 10*). We also show that ORF1p and L1mRNA, probably in a complex with ORF2p, enters the nucleus during mitosis, accumulating in cells in the G1 phase (*Figure 2*). The nuclear localization of L1 RNPs upon transition through mitosis may be due to simple diffusion of L1 RNPs in the nuclear proximity and subsequent sequestration of the particles into the nucleus upon nuclear membrane formation. Another interesting possibility, is that, during mitosis, the L1 RNPs may weakly interact with chromatin, most likely through the positively charged ORF1p. These interactions could increase the chances of L1 RNPs being 'trapped' in the nucleus after reformation of the nuclear membrane. This hypothesis is supported by our unpublished observations that GFP-tagged ORF1p strongly interact with chromatin during metaphase and this process seems to increase the amount of ORF1p observed into the nucleus in early G1. Moreover, as shown in *Figure 8* and *Figure 8—figure supplement 1* of *Figure 8*, we always observe ORF1p in chromatin fractions even in the absence of ORF2p (unpublished data). This observation strongly suggests interaction of ORF1p with chromatin independent of retrotransposition itself.

We observe a CRM1 mediated/leptomycin sensitive nuclear export of ORF1p (*Figure 4*) that keeps the nuclear level of ORF1p low and helps explain the observation that ORF1p is always mostly cytoplasmic (*Figure 1* and *Figure 4*). Future studies will need to explore the CRM1/ORF1p interaction and the role of this interaction in L1 RNP cellular dynamics and retrotransposition. The decoupling of the CRM1 role on the cell cycle from its importance on L1 cellular localization will be challenging but essential for the understanding of ORF1p nuclear export. We also show that even if L1 enters the nucleus in M phase, retrotransposition does not happen during cell division (M phase) but it is during the following S phase in which retrotransposition peaks (*Figures 5–7*). The finding that L1 retrotransposition has a strong bias for S phase is, in retrospect, not entirely surprising considering that deoxynucleoside triphosphates (dNTPs), critically necessary for reverse transcription, are at high levels during the S phase and are greatly restricted during the other cell cycle stages (*Hofer et al., 2012*; *Stillman, 2013*). This layer of metabolic regulation may reflect an ancient adaptation to limit the proliferation of retroelements. dNTP concentration is tightly controlled by ribonucleotide reductase (RNR), the enzyme that converts ribonucleotide diphosphates (rNDPs) into dNDPs and by SAMHD1 (sterile alpha motif and HD-domain containing protein 1) that cleaves dNTPs to deoxynucleosides (*Stillman, 2013*). Indeed, SAMHD1 expression was found to restrict replication of lentiviruses such as HIV, by restricting availability of dNTPs (*Hrecka et al., 2011*; *Goldstone et al., 2011*). It is therefore not surprising that SAMHD1 was also shown to restrict LINE-1 retrotransposition (*Zhao et al., 2013*) directly supporting the idea that dNTP concentration can profoundly limit L1 jumping. Interestingly, we also found that mimosine, a compound that blocks the cell cycle in G1/early S and that inhibits RNR, also strongly inhibits L1 retrotransposition (*Figure 5*). Our findings suggest that inhibition of L1 retrotransposition mediated by mimosine involves multiple mechanisms other than cell cycle inhibition. It is possible that the depletion of dNTPs by mimosine further mediates inhibition of L1 retrotransposition. In complete accord with these observations is our finding that retrotransposition happens in S phase, during which dNTP concentration peaks,

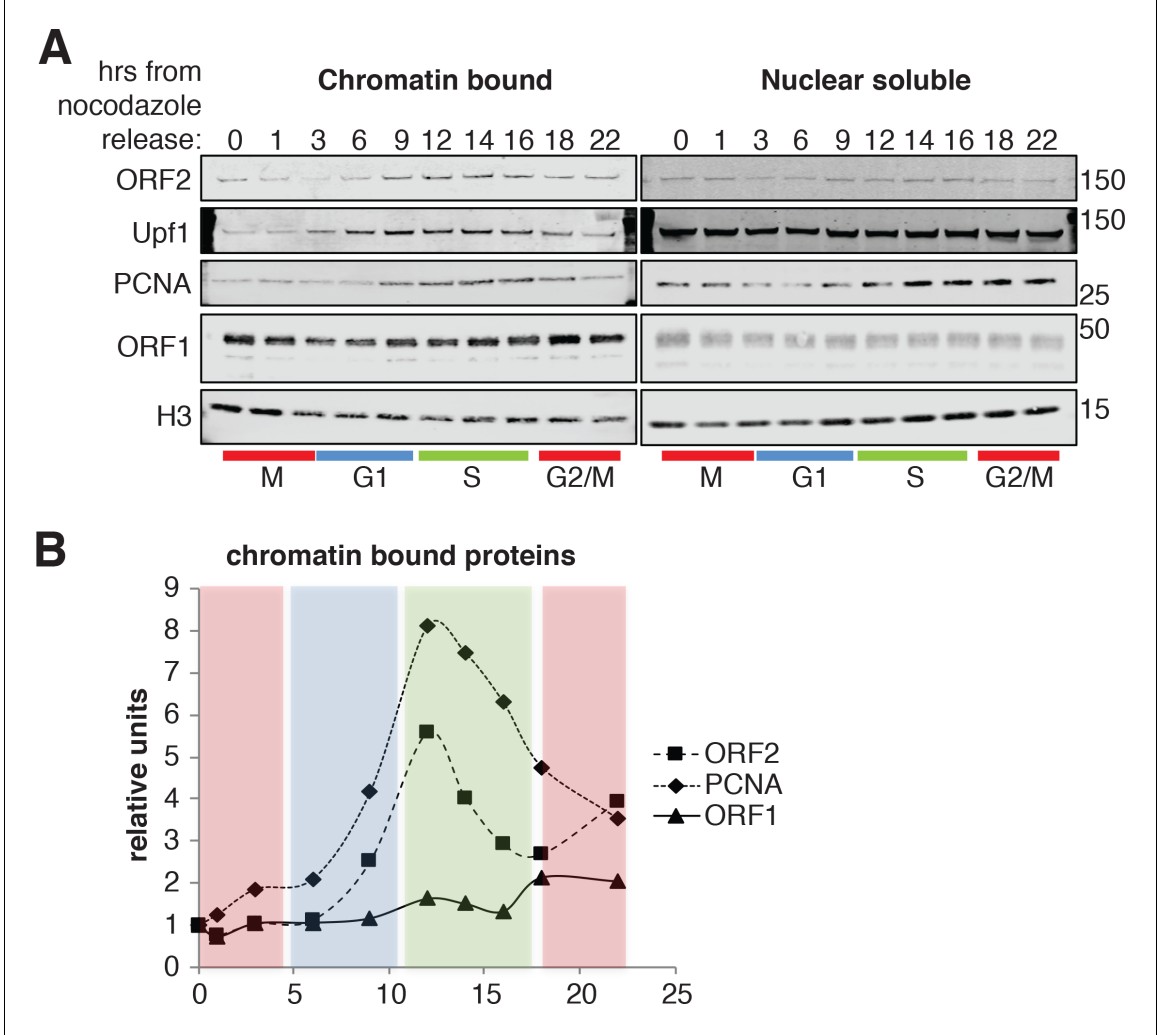

**Figure 8.** ORF2p binds chromatin during S phase. (**A**) Western blot of chromatin bound proteins or nuclear soluble proteins extracted from HeLa M2 cells expressing recoded L1, synchronized with nocodazole and released into the cell cycle for the indicated times. The colored bars below the blots indicate cell cycle phases extrapolated from *Figure 4—figure supplement 1*. The targets of the antibodies used for blotting are indicated. The quantification of the ORF2p, ORF1p and PCNA proteins bound to chromatin are reported in (**B**) as ratio of H3 signals. The relative units at time zero are set to 1.

DOI: https://doi.org/10.7554/eLife.30058.028

The following figure supplement is available for figure 8:

**Figure supplement 1.** Analysis of nuclear fractions.

DOI: https://doi.org/10.7554/eLife.30058.029

allowing efficient reverse transcription and thus L1 retrotransposition. This may thus be viewed as an adaptation of the retroelement to a host defense.

Our functional studies showing S phase bias of L1 retrotransposition are also corroborated by our biochemical observations that show that ORF2p is recruited to chromatin during S phase (*Figure 8*) and suggest that ORF2p is recruited to a subset of sites of DNA replication with PCNA and MCM proteins (*Figure 9*). Mass spectrometry analysis revealed interaction of the previously described PCNA/ORF2p complex (*Taylor et al., 2013*) with TOP1, RPA1 and PARP1 proteins (*Taylor et al., 2018*) all of which associate with replication forks. Interestingly, the PARP1 interaction suggests that L1 specifically interacts with stalled replication forks (*Berti et al., 2013*). Our co-localization of PCNA and ORF2p also supports the presence of ORF2p at potential sites of DNA replication, marked by PCNA staining during the S phase. PCNA and ORF2p immunofluorescence revealed that only some PCNA foci of replication overlap with ORF2p nuclear foci, suggesting that, at least in some

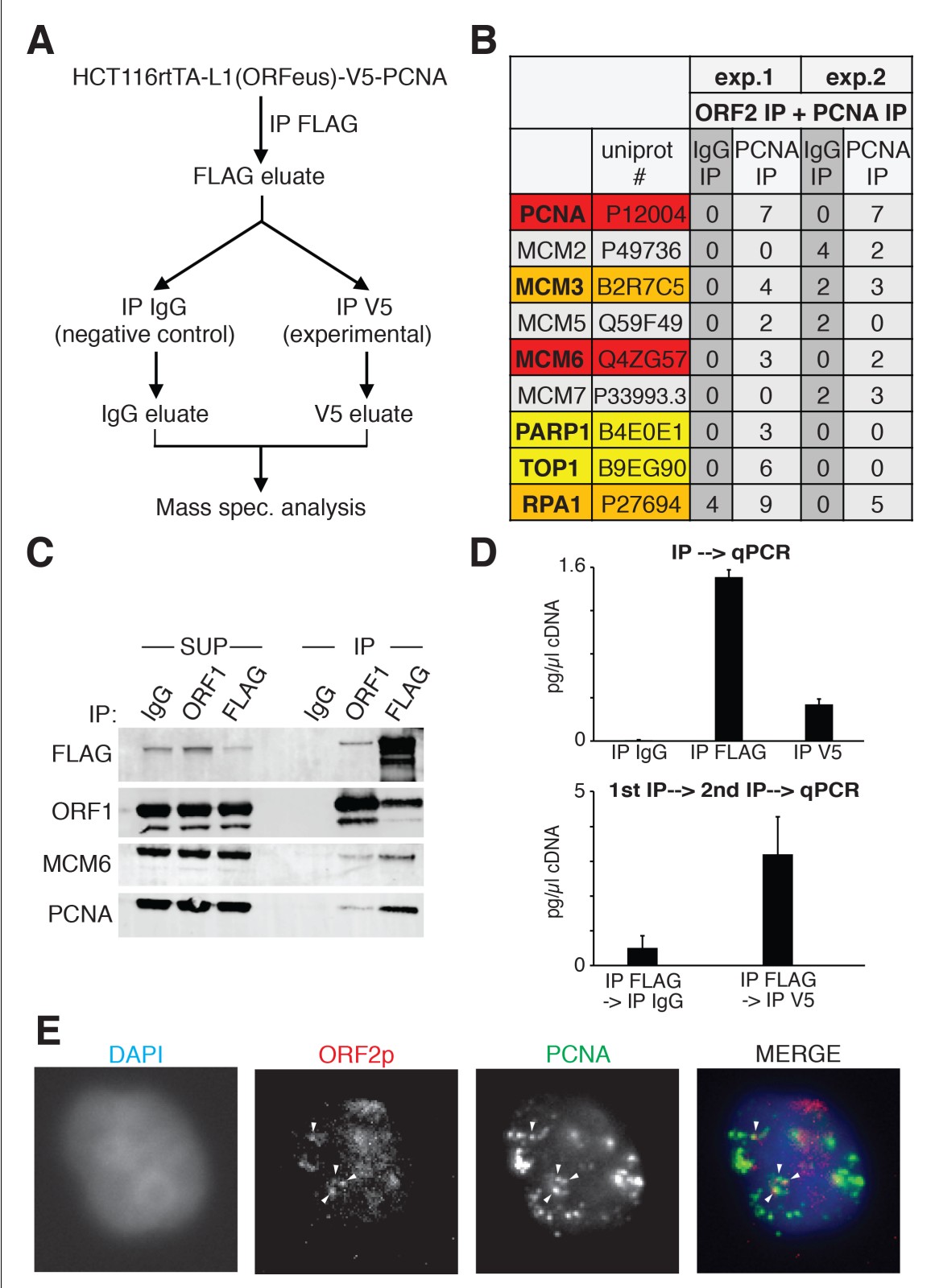

**Figure 9.** ORF2p is associated with replication fork proteins during S phase. (**A**) Schematic of sequential ORF2-PCNA immunoprecipitation and mass spectrometry analysis. HCT116 expressing V5-PCNA, rtTA and a recoded L1 (ORFeus) with flag-tagged ORF2p were used to immune-precipitate ORF2p. The immunoprecipitated complexes eluted with FLAG peptides were split in two and used for immunoprecipitation with V5 or IgG control antibodies. After native elution with V5 peptides the samples were analyzed by mass spectrometry. (**B**) Peptide numbers of known DNA replication fork

*Figure 9 continued on next page*

*Figure 9 continued*

proteins (identified by their UniProt number) obtained after mass spectrometry analysis of the ORF2p-PCNA/IgG sequential IP. Peptide counts of two independent experiments are reported. Red = high confident interactors that are found in both experiments and with no peptides in the IgG control IPs; orange = possible interactors that are found in both experiments and with some peptides in the IgG control IPs; yellow = low confidence interactors that are found in just one experiment and with no peptides in the IgG control IPs; gray = MCM proteins not identified as ORF2p interactors in our analysis. (C) Western blot of MCM6 and PCNA proteins co-immunoprecipitated upon IgG (control), ORF1 or FLAG(ORF2) IP. Grindates of 293T$_{LD}$ cells expressing a recoded L1 with flag-tagged ORF2p were used. (SUP = supernatant after IP, IP = immunoprecipitation). (D) RT-qPCR quantification of L1 after direct immunoprecipitation from HCT116-V5-PCNA-L1 cells using IgG, FLAG-M2 (ORF2p) or V5 (PCNA) antibodies (top panel) or after sequential IP using FLAG-IgG or FLAG-V5 antibodies (bottom panel). "cDNA" amount refers to amount of cDNA produced in in vitro RT reactions and reflects mRNA concentrations in the immunoprecipitates. (E) Immunostaining of ORF2p (red) and PCNA (green) in cells synchronized with double thymidine block and released for 2 hr in S phase.
DOI: https://doi.org/10.7554/eLife.30058.030

The following source data is available for figure 9:

**Source data 1.** Complete list of proteins and peptides identified by mass spectrometry.
DOI: https://doi.org/10.7554/eLife.30058.031

instances, L1, possibly engaged in TPRT, may specifically interact with a subset of perhaps stalled

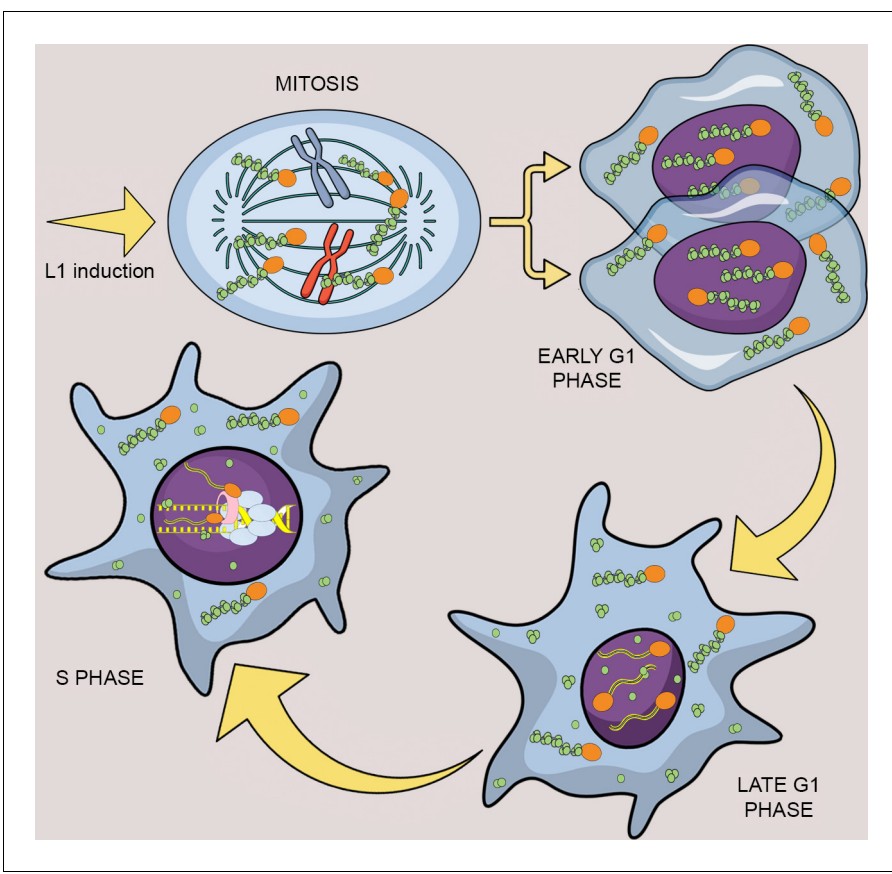

**Figure 10.** Model of L1 protein dynamics and L1 retrotransposition during the cell cycle. After induction of L1 expression, L1 RNPs formed by ORF1p (green balls), ORF2p (orange balls) and L1mRNA (blue/yellow line interacting with ORF2p and ORF1p trimers), enter the nucleus as a result of mitotic nuclear membrane breakdown. In early G1 phase, when the nuclear membrane re-assembles, L1 RNPs are found in the nucleus. ORF1p, but not ORF2p is then exported from the nucleus through a CRM1 dependent mechanism leaving ORF2p and L1 mRNA into the nucleus. During S phase and DNA replication the ORF2p-mRNA L1 particles retrotranspose into new loci of the genome and interact with components of the replication fork: PCNA (pink tube) and MCM proteins (light blue balls).
DOI: https://doi.org/10.7554/eLife.30058.032

replication forks. Our data do not clarify whether the replication fork stall is caused by ORF2p nicking of the DNA or alternatively, whether the retrotransposing L1 complex is recruited specifically to previously stalled replication forks. Conversely, not all ORF2p nuclear foci overlap with PCNA sites, supporting a model in which L1 interaction with the replication fork may represent just one of several modes used by L1 to select a DNA target site and retrotranspose. Our previous work (*Taylor et al., 2013*) showed that PCNA does not interact with ORF2p mutated in its endonuclease (EN⁻) or reverse transcriptase (RT-) domain. These observations led us to hypothesize that PCNA is recruited by ORF2p after the first steps of TPRT (nicking of genomic DNA and beginning of L1 mRNA reverse transcription), and not vice-versa (chromatin recruitment of ORF2p by PCNA). Together with the observations presented in this manuscript we envision a model in which L1 RNPs, comprising at least ORF2p and L1mRNA, are recruited to replication forks in S phase during DNA replication. A subset L1 RNPs subsequently mediate productive retrotransposition into target loci, perhaps aided by stalling of the replication fork. During TPRT, PCNA, readily available at the site of DNA replication, can be recruited in the latter steps of retrotransposition perhaps to mediate repair of the newly synthesized L1 cDNA/genomic DNA junctions. It is intriguing to postulate that ligases involved in DNA replication such as ligase 1 (LIG1) may also help seal the final nicks in L1 retrotransposition events. An alternative model that could, at least partially, explain our data hypothesizes that a replication fork collides with a nicked DNA formed by retrotransposing L1. In this latter model, the co-localization of ORF2p with PCNA and MCM proteins would happen after endonuclease cut and initiation of RT by ORF2p. Future work, most likely based on single molecule observation of the retrotransposing L1 RNPs, will be needed to validate this still speculative models.

Interestingly, our observations on ORF1p cytoplasmic/nuclear dynamic suggest deeper implications. The fact that ORF1p is exported from the nucleus before S phase, leads to the conclusion that retrotransposition, happening mainly during DNA replication, is mediated by RNPs depleted of ORF1p and constituted only or predominantly by ORF2p and L1 mRNA, a conclusion also supported by data presented by Taylor et al. (*Taylor et al., 2018*). In vitro studies of TPRT show that the first and presumably critical steps in retrotransposition can efficiently occur in vitro in the absence of ORF1p (*Cost et al., 2002*). The hypothesis that ORF1p is dispensable for the actual DNA cutting and reverse transcription steps in vivo, is supported by our observation that nuclear ORF1p can be specifically recognized by one of our antibodies (JH73g). We interpret this observation to mean that this antibody recognizes a specific conformational state of ORF1p unique to the nucleus. Not surprisingly, the nuclear form of ORF1p that is recognized by the JH73g Ab, has impaired binding to ORF2p, suggesting that once inside the nucleus, ORF1p may dissociate from the L1 RNPs destined to carry out the critical endonuclease/reverse transcription steps of retrotransposition during S phase. Moreover, most of the ORF1p does not interact with PCNA and MCM6 that, instead, interact mainly with ORF2p. We also observed (rare) instances of cells clearly expressing ORF2p in the nucleus in the absence of detectable ORF1p (*Figure 1* and (*Taylor et al., 2018*)). Finally, observations of HeLa.S-FUCCI cells expressing Halo tagged ORF1p show that ORF1p is never nuclear in cells in the S/G2/M phase. Halo tagged ORF2p, in contrast, was observed in the nucleus of certain cells in S/G2/M (*Figure 2—figure supplement 2*) suggesting that, during these cell cycle stages, ORF2p is in the nucleus without ORF1p. Overall, these data suggest that chromatin bound and retrotransposition-competent L1 particles are depleted of ORF1p and mainly consist of ORF2p in complex with L1 mRNA and host factors involved in retrotransposition. Future studies are necessary to better understand the differences of nuclear and cytoplasmic ORF1p and the molecular processes that may mediate ORF1p depletion from L1 RNPs. An attractive possibility is that ORF1p dissociation from L1 RNPs might be associated with delivering an ORF2-RNA RNP to chromatin, although we do not have direct evidence for this. It is tempting to imagine that in the nucleus, the absence of ORF1p trimers, thought to bind L1 mRNA every 50 nucleotides in the cytoplasm (*Khazina et al., 2011*), may promote ORF2p's unhindered movement during reverse transcription of L1 mRNA in the process of TPRT. This hypothesis will need to be better explored in future studies, by examining ORF2p and L1 mRNA dynamics during the cell cycle. The lack of a sensitive Ab against ORF2p, the fact that most cells expressing ORF1p do not express ORF2p due to an unknown post-transcriptional mechanism controlling ORF2p expression (*Taylor et al., 2013*; *Alisch et al., 2006*; *Luke et al., 2013*) and the difficulties of detecting ORF2p even in the context of overexpression (*Doucet et al., 2016*), continue to technically challenge the study of L1 cellular dynamics.

More recent advances in the study of L1 retrotransposon, such as the construction and characterization of ORFeus with its increased expression and function (*An et al., 2011*; *Han and Boeke, 2004*), the use of smaller and brighter fluorescent tags that allow the exploration of the temporal axis of retrotransposition and the implementation of sensitive biochemical approaches (*Sakaue-Sawano et al., 2008*; *Subach et al., 2009*; *Grimm et al., 2015*) enabled us to discover new and unexpected interactions between L1 and the 'host' cell. It is not surprising that retrotransposons, evolved within the human genome for millions of years, have 'learned' to leverage important cellular pathways, such as the cell cycle and DNA replication, for their own purpose of spreading and increasing their genomic content (*Boissinot et al., 2000*; *Boissinot and Sookdeo, 2016*). On the other hand, it is also increasingly clear how cells respond to L1 expansions during evolution, engaging in an ongoing genetic arms-race (*Daugherty and Malik, 2012*; *Molaro and Malik, 2016*). For example, as previously proposed, the nuclear membrane may have represented one of many barrier that retrotransposons had to overcome to maintain effective retrotransposition frequency (*Boeke, 2003*; *Koonin, 2006*).

## Materials and methods

### Cell lines

HeLa M2 cells (a gift from Gerald Schumann, Paul-Ehrlich-Institute; (*Hampf and Gossen, 2007*) were cultured in DMEM media supplemented with 10% FBS (Gemini, prod. number 100–106) and 1 mM L-glutamine (ThermoFisher/Life Technologies, prod. number 25030–081) (complete medium). Cells were routinely split in fresh medium upon reaching 80–90% confluency. During routine culture of the cells the medium was changed every 2–3 days.

293T$_{LD}$ cells adapted to suspension (*Taylor et al., 2013*) were used for transfection with PEI and collected to generate cell grindates as previously described in (*Taylor et al., 2013*). Cell grindates were used for IPs presented in *Figure 9C*.

HCT116 colorectal carcinoma cells were cultured in McCoy's 5A media (Life Technologies/Gibco, prod. number 16600–108) supplemented with 10% FBS and 1 mM L-glutamine. A subline of these cell line and expressing a V5-PCNA protein was generated using a CRISPR approach. A single Cas9-gRNA vector (PM192) was generated by Golden Gate reaction (*Ran et al., 2013*) using vector pX459V2.0 (a gift from Feng Zhang, Addgene, plasmid number 62988) and an annealed DNA duplex:

JB17486-F2: CACC G GTCTAGCTGGTTTCGGCTTC
JB17487-R2: Aaac GAAGCCGAAACCAGCTAGAC C
A gBlock DNA purchased from IDT integrated DNA technologies was used as donor DNA.
PM-GB3: ccgtgggctggacagcgtggtgacgtcgcaacgcggcgcagggtgagagcgcgcgcttgcggacgcggcgg-cattaaacggttgcaggcgtagcagagtggtcgttgtctttctagGTCTCAGCCGGTCGTCGCGACGTTCGCCCGC TCGCTCTGAGGCTCGTGAAGCCGAAACCAGCTAGACTTTCCTCCTTCCCGCCTGCCTG TAGCGGCGTTGTTGCCACTCCGCCACCATG GGT AAG CCT ATC CCTAACCCTCTCCTCGGTCTC GATTCTACGGGAGAAGGGCAAGGGCAAGGGCAAGGGCCGGGCCGCGGCTACGCGTATCGA TCCTTCGAGGCGCGCCTGGTCCAGGGCTCCATCCTCAAGAAGGTGTTGGAGGCACTCAAGGACC TCATCAACGAGGCCTGCTGGGATATTAGCTCCAGCGGTGTAAACCTGCAGAGCATGGACTCG TCCCACGTCTCTTTGGTGCAGCTCACCCTGCGGTCTG

The donor DNA was transfected together with PM192 plasmid using Fugene-HD reagent (Promega, Madison, WI; prod. number E2311) in HCT116 plated in a six well plate. Cells were transfected with 200 ng of donor DNA, 1.5 μg PM192 and 6.8 μl Fugene-HD in 100 μl Opti-MEM (Thermo Fisher scientific, prod. number 31985088). 24 hr after transfection, cells were selected in 1 ug/ml puromycin for additional 48 hr. Single clones were picked after serial dilution of the cells in complete media without puromycin. Clones were screened in 96 well plates for expression of V5 by immunofluorescence staining. The positive clones were then validated by genomic DNA PCR using primers flanking the site of CRISPR cut. The amplified band was gel isolated and sequenced by Sanger sequencing. The primers used are the following:

JB17488-PCNACseqF
CTGCAGATGTACCCCTTGgt
JB17489-PCNACseqR

GACCAGATCTGACTTTGGACTT

The positive clones were then also validated by western blotting analysis using an antibody against PCNA and V5 tag.

The HCT116-V5-PCNA cell line was then used to generate stable cell lines expressing the rtTA transactivator. The pTet-ON advance vector (Clonetech, prod. number 631069) was transfected and cells were then selected for several weeks using media supplemented with 250 µg/ml neomycin. Clones were screened using a construct encoding GFP under the control of a tetracycline activated promoter. Finally, HCT116-V5-PCNA-rtTA cell lines stably maintaining episomal pCEP-puro-plasmids expressing ORFeus (LD401) were generated and cultured under puromycin selection (1 µg/ml) to prevent the loss of the L1 plasmids and neomycin (250 µg/ml) to prevent the loss of rtTA transactivator.

120 15 cm plates of HCT116-V5-PCNA-rtTA-LD401 cells were used to immunoprecipitate ORF2p-Flag using Flag-M2 antibodies (Sigma, prod. number F1804). The immunoprecipitated protein complexes were then used as input to immunoprecipitate V5-PCNA with V5 antibodies (Invitrogen, prod. number 46–1157). Immunoprecipitation assays were conducted as described below.

HeLa.S-Fucci cells were purchased from the Riken BRC cell bank (RIKEN BioResource Center, Japan, prod. number RCB 2812). Stable HeLa.S-FUCCI cells expressing rtTA were generated transfecting a pTet-ON advance vector (Clontech, prod. number 631069) subcloned to carry a blasticidin resistance cassette instead of a neomycin resistance cassette. Cells were selected for several weeks in 15 µg/ml blasticidin and several clones screened for the expression of firefly luciferase under a control of a doxycycline promoter (gift from S.K. Logan laboratory). The selected stable HeLa.S-FUCCI-rtTA cell lines were cultured in complete DMEM media supplemented with 15 ug/ml blasticidin.

## Plasmids and DNA constructs

LD401 (ORFeus with 3xFlag ORF2) and MT302 (L1rp with 3xFlag ORF2 and L1 5'UTR) plasmids were previously described and characterized in (Taylor et al., 2013).

PM160 (ORfeus with L1rp 5'UTR) plasmid was obtained through Gibson reaction of a PCR L1 5' UTR DNA fragment from MT302 used as template.

PM226 (L1rp with 3xFlag ORF2 and without L1 5'UTR) was generated by ligation of a synthetic 5' end of L1rp without UTR to the BsiWI-PmlI cut MT302 vector.

EA79 (untagged ORFeus with GFP-AI cassette) was constructed subcloning ORFeus under a Tet inducible promoter into a pCEP-4 plasmid (ThermoFisher scientific, prod. number V04450) in which the hygromycin resistance cassette was substituted with a puromycin resistance cassette, as previously described in (Taylor et al., 2013) (pCEP-puro plasmid).

The fluorescent timer FT-AI cassette was built with Gibson assembly using two synthetic DNA fragments (Quinglan Biotech and Twist Bioscience) ligated into the BstZ17I 3'UTR of untagged ORFeus (PM260). Sequences of the fast fluorescent timer mCherry variant are the same of plasmid pTRE-Fast-FT (a gift from Vladislav Verkhusha, Addgene plasmid number 31913)(Subach et al., 2009).

Constructs expressing Halotag7-ORF1p (PM285) and Halotag7-ORF2p (PM283) were made by Gibson assembly of a PCR DNA fragment encoding the HaloTag7 (pLH1197-pcDNA3.1-PfV-Halo, a gift from Liam Holt) and pCEP-puro-ORFeus vector. The tag was inserted right before ORF1p and ORF2p stop codon downstream of a G4S linker.

The MEK1-GFP expression plasmid was purchased from Addgene (plasmid #14746).

All constructs were verified by Sanger sequencing (Genewiz).

## Immunoprecipitation, electrophoresis and western analysis

HeLa M2 cells were lysed in SKL Triton lysis buffer (50 mM Hepes pH7.5, 150 mM NaCl, 1 mM EDTA, 1 mM EGTA, 10% glycerol, 1% Triton X-100, 25 mM NaF, 10 µM ZnCl$_2$) supplemented with protease and phosphatase inhibitors (Complete-EDTA free, Roche/Sigma prod. number 11873580001; 1 mM PMSF and 1 mM NaVO$_4$). NuPage 4XLDS sample buffer (ThermoFisher Scientific, prod. number NP0007) supplemented with 1.43M β-mercaptoethanol was added to the samples to reach a 1X dilution (350 mM β-mercaptoethanol final concentration) before gel electrophoresis performed using 4–12% Bis-Tris gels (ThermoFisher Scientific, prod. number

WG1402BOX). Proteins were transferred on Immobilon-FL membrane (Millipore, prod. number IPFL00010), blocked for 1 hr with blocking buffer (LiCOR prod. number 927–40000):TBS buffer (50 mM Tris Base, 154 mM NaCl) 1:1 and then incubated with primary antibodies solubilized in LiCOR blocking buffer:TBS-Tween (0.1% Tween in TBS buffer) 1:1. Secondary donkey anti-goat antibodies conjugated to IRDye680 (anti-rabbit) or IRDye800 (anti-mouse) dyes (LiCOR prod. number 926–32210 and 926–68071), were used for detection of the specific bands on an Odyssey CLx scanner (LiCOR). 293T$_{LD}$ grindates used in *Figure 9C* were lysed in extraction buffer (20 mM Hepes pH 7.4, 500 mM NaCl, 1% Triton X-100) supplemented with protease and phosphatase inhibitors.

Immunoprecipitations were performed using 5–10 µl of dynabeads conjugated to primary antibodies (Dynabeads Antibody Coupling kit, Life Technologies, prod. number 14311D) incubated with lysates for at least 1 hr nutating at 4°C. After five washes in lysis buffer the immunocomplexes were eluted either in sample buffer, shaken with beads for 10 min at 70°C or using 3xFLAG (Sigma, prod. number F4799) or V5 (Sigma, prod. number V7754) peptides shaken with beads for 1 hr at 4°C. After elution, supernatants were collected and β-mercaptoethanol added to a final concentration of 350 mM.

Antibodies against ORF1p used in this study are:

- 4H1 = Mouse monoclonal antibody targeting amino acids 35 to 44 of human ORF1p (*Rodić et al., 2014*; *Taylor et al., 2013*; *Doucet-O'Hare et al., 2015*) (Millipore, cat. number MABC1152).
- JH74 = Rabbit monoclonal antibody raised against the C-terminus of human ORF1p (*Doucet-O'Hare et al., 2015*) (a gift from Dr. Jeffry Han)
- JH73g = Rabbit monoclonal antibody raised against the C-terminus of human ORF1p and immuno-purified by Genscript from hybridoma cells (a gift from Dr. Jeffry Han). Please note that this antibody is distinct from JH73 (*Taylor et al., 2013*) (see *Figure 4—figure supplement 1*).
- 4632 = Rabbit polyclonal antibody against human ORF1p (a gift from Dr. Thomas Fanning)

We also compared the previously described JH73 antibody (*Taylor et al., 2013*) with the JH73g antibody (*Figure 4—figure supplement 1*). These two antibodies were derived from the same rabbit immunization using a purified globular ORF1p C-terminus (Dr. Jeffry Han, unpublished data). We confirmed that JH73g Ab is distinct from JH73 Ab displaying clear differences in the light chain migration and base peak chromatogram (*Figure 4—figure supplement 1*).

## IP-RT-qPCR

Immunoprecipitation of L1 complexes used for the analysis of L1 mRNA were conducted exactly as for the IPs used for mass spectrometry analysis of ORF2p-PCNA complexes. It is worth noting that the buffer used to IP the ORF2p-PCNA complex (SKL Triton lysis buffer) is different from the elution buffer (EB500) used in (*Taylor et al., 2013*) and optimized for the detection of ORF1p and ORF2p interaction. We could detect a much lower amount of PCNA interacting with ORF2p using EB500 buffer and we were not able to detect L1 mRNA after sequential ORF2p and PCNA IP, using EB500 buffer.

HCT116-V5-PCNA expressing ORFeus (LD401) (about $1 \times 10^6$ cells for single IP and about $70 \times 10^6$ cells for sequential IP) were collected by trypsinization and lysed in SKL Triton lysis buffer (50 mM Hepes pH7.5, 150 mM NaCl, 1 mM EDTA, 1 mM EGTA, 10% glycerol, 1% Triton X-100, 25 mM NaF, 10 µM ZnCl$_2$) supplemented with 1 mM DTT, 400 µM Ribonucleoside Vanadyl Complex (VRC) (NEB, prod. number S1402), 400U per ml of buffer of RNASEOUT (Thermo, prod. number 10777019), protein inhibitor and phosphatase inhibitor (Complete-EDTA free, Roche/Sigma prod. number 11873580001; 1 mM PMSF and 1 mM NaVO$_4$). The cells-lysate mixture was stored at −80°C over-night. The next day the mixture was thawed on ice and centrifuged for 15 min at 16,000 rcf at 4°C. ORF2p complexes were immunoprecipitated over-night at 4°C using a mouse FLAG-M2 antibody (Sigma, pred. number F1804) coupled to magnetic beads (Dynabeads Antibody Coupling kit, Life Technologies, prod. number 14311D). Beads were washed three times in Triton buffer and protein complexes were then eluted using 100 µl of 1 mg/ml 3xFLAG peptide (Sigma, product. Number F4799) supplemented with protease, phosphatase and RNase inhibitors for 1 hr at 4°C under shaking. The 3xFLAG eluate was split into half and used for the second immunoprecipition using normal mouse IgG control (Santa Cruz, prod. number sc2025) or V5 (Invitrogen, prod. number 46–1157)

coupled beads for 4 hr. Beads were washed three times in Triton buffer, resuspended in 100 µl of Triton buffer containing 30 µg of proteinase K (Invitrogen, prod. number 25530049). The mixture was incubated at 55°C for 30 min. 1 ml of Trizol (Life Technologies, prod. number 15596026) was directly added to the beads mixture and mRNA was purified using RNA clean-up and concentration columns (Norgen Biotek, prod. number 23600). cDNA was generated from RNA (380 ng for single IP and 245 ng for sequential IP) using the USB First-Strand cDNA synthesis kit for Real-Time PCR (Affimetrix, prod. number 75780). Q-PCR was performed using a standard curve of LD401 plasmid. Each Q-PCR reaction contained 2.5 µl of Sybr Green mastermix 2X (Roche, LightCycler 480 SYBR Green I Master, prod. number 04887352001), 25 nl of forward primer (100 µM), 25 nl of reverse primer (100 µM), 100 nl of cDNA for the single IP and 500 nl of cDNA for the sequential IP and water to 5 µl (final volume). Q-PCR was performed using a Light Cycler 480 (Roche) with standard conditions. The primers used for qPCR amplify a 185 bp amplicon in ORF1 and are reported below:

JB13415 (forward): GCTGGATGGAGAACGACTTC
JB13416 (reverse): TTCAGCTCCATCAGCTCCTT

## Mass spectrometry analysis

Samples were reduced and alkylated with DTT (1 hr at 57°C) and iodoacetamide (45 min at room temperature). The samples were then loaded on a NuPAGE 4–12% Bis-Tris gel (ThermoFisher Scientific, prod. number WG1402BOX) and ran for only 10 min at 200V. The gel was then stained with GelCode Blue Staining Reagent (ThermoFisher, prod. number 24590) and the protein bands were excised. The gel plugs were cut into 1 mm$^3$ pieces, washed and destained with 1:1 (v/v) (methanol:100 mM ammonium bicarbonate). After at least five solvent exchanges the gel plug was dehydrated by aspirating 100 µl of acetonitrile (ACN) and further dried in the SpeedVac. In-gel digestion was performed by adding 250 ng of trypsin (ThermoFisher, prod Number 90057) onto the dried gel plug followed by 300 µl of ammonium bicarbonate 100 mM. Digestion was carried out overnight at room temperature with gentle shaking. The digestion was stopped by adding 300 µl of R2 50 µM Poros beads in 5% formic acid and 0.2% trifluoro acetic acid (TFA) and agitated for 2 hr at 4°C. Beads were loaded onto equilibrated C18 ziptips and additional aliquots of 0.1% TFA were added to the gel pieces and the wash solution also added to the ziptip. The Poros beads were washed with additional three aliquots of 0.5% acetic acid and peptides were eluted with 40% acetonitrile in 0.5% acetic acid followed by 80% acetonitrile in 0.5% acetic acid. The organic solvent was removed using a SpeedVac concentrator and the samples reconstituted in 0.5% acetic acid. An aliquot of each sample was loaded onto an Acclaim PepMap100 C18 75 µm x 15 cm column with 3 µm bead size coupled to an EASY-Spray 75 µm x 50 cm PepMap C18 analytical HPLC column with a 2 µm bead size using the auto sampler of an EASY-nLC 1000 HPLC (Thermo Fisher Scientific) and solvent A (2% acetonitrile, 0.5% acetic acid). The peptides were eluted into a Thermo Fisher Scientific Orbitrap Fusion Lumos Tribrid Mass Spectrometer increasing from 5% to 35% solvent B (80% acetonitrile, 0.5% acetic acid) over 60 m, followed by an increase from 35% to 45% solvent B over 10 m and 45–100% solvent B in 10 m. Full MS spectra were obtained with a resolution of 120,000 at 200 m/z, an AGC target of 400,000, with a maximum ion time of 50 ms, and a scan range from 400 to 1500 m/z. The MS/MS spectra were recorded in the ion trap, with an AGC target of 10,000, maximum ion time of 60 ms, one microscan, 2 m/z isolation window, and Normalized Collision Energy (NCE) of 32. All acquired MS2 spectra were searched against a UniProt human database using Sequest within Proteome Discoverer (Thermo Fisher Scientific). The search parameters were as follows: precursor mass tolerance ±10 ppm, fragment mass tolerance ±0.4 Da, digestion parameters trypsin allowing two missed cleavages, fixed modification of carbamidomethyl on cysteine, variable modification of oxidation on methionine, and variable modification of deamidation on glutamine and asparagine and a 1% peptide and protein FDR searched against a decoy database. The results were filtered to only include proteins identified by at least two unique peptides.

Proteins identified exclusively in the V5-PCNA IP (279 in analysis 1 and 158 in analysis 2) and not in the parallel IP with the control IgG were considered components of the PCNA-ORF2 complex (*Figure 9* and *Figure 9—source data 1*). These proteins were queried in the STRING database (*Szklarczyk et al., 2015*) to identify known interactions between these candidates. The most enriched GO Biological Process classes were RNA splicing (GO ID = 0008380; FDR = 1.18e-14) for the first analysis and viral process (GO ID = 0016032; FDR = 1.05e-16) for the second analysis.

Mass spectrometry analysis of JH73 and JH73g antibodies was performed as follows: the antibodies were buffer exchanged to 100 mM ammonium bicarbonate using 7K molecular weight cutoff Zeba Spin Desalting columns (Thermo Scientific) as per the manufacturer recommended protocol to remove glycerol. Following buffer exchange, samples were denatured with 8M Urea in Tris-HCl solution. Denatured samples were reduced with dithiothreitol (2 µl of a 1 M solution) at 37°C for 1 hr and then alkylated with iodoacetic acid at room temperature in dark for 45 min (12 µl of a 1M solution). Samples were diluted to final urea concentration of 2M to facilitate enzymatic digestion. Each sample was split into two aliquots and digested using trypsin and pepsin. For trypsin digestion, 100 ng of sequencing grade-modified trypsin was added and digestion proceeded overnight on a shaker at RT. To inactivate the trypsin, samples were acidified using trifluoroacedic acid (TFA) to final concentration of 0.2%. For pepsin digest, samples were acidified to pH <2 using 1M HCl and digested using 100 ng of pepsin (Promega) for 1 hr at 37°C. Pepsin was heat inactivated by incubating the samples at 95°C for 15 min. Peptide extraction was performed by addition of 5 µl of R2 20 µm Poros beads slurry (Life Technologies Corporation) to each sample. Samples were incubated with agitation at 4°C for 3 hr. Peptide extraction was performed as described above. Digested peptides were loaded onto the column using the HPLC set up described above. The peptides were gradient eluted directly into a Q Exactive (Thermo Scientific) mass spectrometer using a 30 min gradient from 5% to 30% solvent B (80% acetonitrile, 0.5% acetic acid), followed by 10 min from 30% to 40% solvent B and 10 min from 40% to 100% solvent B. The Q Exactive mass spectrometer acquired high resolution full MS spectra with a resolution of 70,000, an AGC target of 1e6, maximum ion time of 120 ms, and a scan range of 400 to 1500 m/z. Following each full MS twenty data-dependent high resolution HCD MS/MS spectra were acquired using a resolution of 17,500, AGC target of 5e4, maximum ion time of 120 ms, one microscan, 2 m/z isolation window, fixed first mass of 150 m/z, Normalized Collision Energy (NCE) of 27, and dynamic exclusion of 15 seconds.

## Immunofluorescence staining, live imaging and nuclear/cytoplasmic fractionation

For immunofluorescence staining, cells were grown on coverslips or chamber-slides (Nunc, prod. number 154534) coated with 10 µg/ml fibronectin (ThermoFisher, prod. number 33016–015) in PBS for 4 hours-over night at 37°C. After plating cells were treated with 0.1 µg/ml doxycycline to induce expression of L1 (ORFeus or L1rp). After induction cells were prefixed adding formaldehyde (Fisher, prod. number F79-500) 11% directly to the culture media to a final concentration of 1%. After 10 min at room temperature the media/formaldehyde mixture was discarded and cells were fixed for 10 min at room temperature with formaline 4% in PBS (Life Technologies, prod. number 10010–049). For PCNA/ORF2p immunofluorescence presented in *Figure 9* cells were fixed for 20 m in cold methanol. Cells were then washed twice in PBS supplemented with 10 mM glycine and three times in PBS. Cells were then incubated for at least 1 hr at room temperature in LiCOR blocking buffer (LiCOR prod. number 927–40000). Upon blocking, cells were incubated over night at 4°C with primary antibodies diluted in LiCOR blocking buffer. The next day cells were washed five times in PBS with 0.1% Triton-X 100 and then incubated in secondary antibodies (Invitrogen, prod. number A11029, A11031, A11034, A11036, A32733) for 1 hr in the dark at room temperature. Cells were then washed five times in PBS with 0.1% Triton-X100 and 3 times in PBS and then coverslips or chamber slides mounted using VectorShield mounting media with DAPI (Vectorlab, prod. number H1200). Pictures were taken using an EVOS-FL Auto cell imaging system (Invitrogen).

Pictures of HeLa-S.FUCCI live cells expressing Halotag7 ORF1p and ORF2p were obtained incubating the cells for 15 min with 100 nM JF646 dye (a gift from Timothee Lionnet)(*Grimm et al., 2015*) in complete FluoroBrite DMEM media (ThermoFisher, prod. number A1896701). After incubation with the dye, the cells were washed twice in PBS before observation under the microscope. Live cell imaging was performed using an EVOS-FL auto cell imaging system with on stage incubator. Live cell images and Z stack movies were obtained using an Andor Yokogawa CSU-x spinning disk on a Nikon TI Eclipse microscope and were recorded with an scMOS (Prime95B, Photometrics) camera with a 100x objective (pixel size 0.11 µM). Images were acquired using Nikon Elements software and analyzed using ImageJ/Fiji (*Schindelin et al., 2012*).

Cytoplasmic/nuclear fractionation was performed using a protein fractionation kit (Thermo prod. number 78840).

## Cell synchronization

HeLa M2 cells were synchronized in M phase by nocodazole treatment and mitotic shake off. Briefly, cells were treated for 12 hr with nocodazole 60 ng/ml and mitotic cells collected in the supernatant after vigorous tapping of the plate. Cells were then washed three times in complete media and released into the cell cycle in fresh complete DMEM media.

HeLa cells were synchronized in G1/S boundary by double thymidine synchronization. $0.35 \times 10^6$ cells were plated in 10 cm culture plates in complete media. After six hours media was exchanged with complete DMEM supplemented with 2 mM thymidine freshly solubilized. After 18 hr, cells were washed three times in PBS and released into the cell cycle with complete media. After 9 hr, 2 mM thymidine medium was added a second time to the cells for additional 15 hr. Cells were then trypsinized, washed twice in PBS and released into the cell cycle in complete DMEM media.

## Propidium Iodide staining and cell cycle analysis

About $1 \times 10^6$ cells were collected in a 1.5 ml tube and fixed at $-20°C$ with 70% cold ethanol for at least 24 hr. Cells were then pelleted by centrifugation at 420 rcf for 5 m and resuspended in HBSS: Phosphate citrate buffer 1:3 (phosphate citrate buffer: 24 parts 0.2M $Na_2HPO_4$ and 1 part of 0.1M citric acid) supplemented with 0.1% Triton X-100. Cells were incubated in this buffer for 30 m at room temperature, subsequently pelleted and resuspended in propidium iodide (PI) staining buffer (20 µg/ml PI, 0.5 mM EDTA, 0.5% NP40, 0.2 mg/ml RNase A in PBS). Cells were incubated in PI staining buffer for 2 hr at 37°C in the dark. After incubation, PI fluorescence was analyzed on an Accuri C6 flow cytometer (BD bioscience) to determine the percentage of cells in each stage of the cell cycle.

## Fluorescence In Situ hybridization (RNA-FISH)

RNA-FISH against recoded and non-recoded L1 was performed using 48 probes against L1 sequence and labelled with cy5 or cy3 fluorophores. The 20 nucleotides probes were designed using Stellaris probe designer from LGC Biosearch Technologies (minimum spacing length = 2; genomic mask factor = 2). The nucleotide sequence of the probes against L1rp and ORFeus is listed in *Supplementary file 1*.

In each well of a 24 well plate, $0.03 \times 10^{\wedge}6$ HeLa-M2 cells expressing ORFeus or L1rp were plated on coverslips coated with 10 µg/ml fibronectin (ThermoFisher, prod. number 33016–015). Expression of L1 was induced with 0.1 µg/ml doxycycline for 24 hr. After induction cells were prefixed adding formaldehyde (Fisher, prod. number F79-500) 11% directly to the culture media to a final concentration of 1%. After 10 min at room temperature the media/formaldehyde mixture was discarded and cells were fixed for 10 min at room temperature with formaline 4% in PBS (Life Technologies, prod. number 10010–049). Cells were then permeabilized for 10 min in PBS supplemented with 0.5% Triton-X 100 followed by a wash in PBS for additional 10 min. Cells were then incubated in pre-hybridization buffer (10% deionized formamide in 2X saline-sodium citrate (SSC) buffer). L1 mRNA was then stained for 3 hr incubating the coverslips in hybridization buffer (for 6 50 µl reactions (total volume = 300 µl): 30 µl of deionized formamide, 6 µl of competitors (5 mg/ml E. coli tRNA +5 mg/ml salmon sperm ssDNA), 1.5 µl of 60 ng/µl probe, 150µl of 20% dextran sulfate, 30 µl of 20 mg/ml BSA, 52.5 µl of water) at 37°C in the dark. After hybridization, coverslips were washed once for 20 min at 37°C in the dark and once at room temperature on a slow shaker with 10% formamide in 2X SSC buffer. Coverslips were then washed for 10 min in PBS, stained with DAPI (0.5 µg/ml in PBS) and mounted on slides using ProLong Gold mounting media (ThermoFisher, prod. number P36930). We observed a much higher nuclear background signal using L1rp probes compared to using ORFeus probes in L1 non-expressing cells. This high background may be explained with non-specific binding of the probes to genomic L1 sequences.

Immunofluorescence staining of ORF1p and ORF2p/Flag was performed after RNA-FISH as reported above.

## Retrotransposition assay

Retrotransposition assays shown in *Figure 5* were performed using HeLa-M2 cells stably expressing plasmid EA79 (pCEP-puro-ORFeus-GFP-AI). The experimental design is reported in *Figure 5A*. Briefly, HeLa M2 cells transfected with EA79 plasmid were selected for at least 5 days in medium

containing 1 µg/ml puromycin to generate stable cell lines that maintain episomal EA79. Expression of recoded L1 was induced with 1 µg/ml doxycycline. Measurements of GFP positive cells were done collecting cells by shake off (*Figure 5B*) or trypsinizing the cells (*Figure 5C and D*) and resuspending them in FACS buffer (HBSS buffer supplemented with 1% FBS, 1 mM EDTA and 100 U/ml of Penicillin-Streptomycin). To exclude dead cells from the measurement of GFP$^+$ cells, 5 µg/ml propidium iodide (PI) was added for at least 5 min to the solution containing cells before measuring the percentage of GFP$^+$ cells with an Accuri C6 flow cytometer (BD bioscience). GFP and PI signals were compensated before analysis. Retrotransposition assays presented in *Figure 6C–D* were performed using HeLa M2 cells transfected with EA79 plasmid and always selected for 5 days in media containing 1 µg/ml puromycin as specified in the experimental design reported in *Figure 6A*.

## Cell cycle analysis of retrotransposition using FT-AI reporter

Measurements of the cell cycle stage of cells that underwent retrotransposition using a fluorescent timer-AI reporter (PM260 plasmid) reported in *Figure 7B and C* were done following two different approaches:

1. Reported in *Figure 7B* and *Figure 7—figure supplement 2A*-B= HeLa M2 cells transfected with plasmid PM260 and selected for 5 days in puromycin were treated with doxycycline 1 µg/ml. After 24 hr of doxycycline induction, blue$^+$red$^-$ and blue$^-$red$^-$ (negative control) cells were sorted using a Sony SH800 sorter (Sony biotechnology Inc.). Cells were sorted directly into a tube containing cold 75% ethanol. The sorted cells were then stained with PI and their cell cycle stage determined as reported above.
2. Reported in *Figure 7C* and *Figure 7—figure supplement 2C*-D= HeLa M2 cells transfected with plasmid PM260 and selected for 5 days in puromycin were treated with doxycycline 1 µg/ml. After 24 hr of doxycycline induction, cells were incubated with complete media containing SYTO61 DNA binding dye (ThermoFisher, prod. number S11343) and doxycycline 1 µg/ml for one hour. Cells were then washed twice in PBS, trypsinized and resuspended in FACS buffer (see above). Fluorescence emissions from the fluorescent timer-AI reporter and SYTO61 DNA dye were measured using an LSRII UV analyzer flow cytometer (BD Bioscience). The cell cycle stage of blue$^-$red$^-$ (negative control), blue$^+$red$^{lower}$ and blue$^+$red$^{higher}$ cells was determined.

Analysis of the percentage of cells in the different cell cycle stages was determined using FlowJo v10.2 software.

## Western blot quantification

Quantification of western blot protein bands was performed using Image Studio ver. 3.1 software on LiCOR CLx scanned images.

## Quantification of ORF1p and ORF2p cellular localization

Quantification of cellular localization of ORF1p and ORF2p was performed counting more than 1000 cells in three different cell preparations. For each experiment, LED intensity, camera gain and exposure were kept constant and set so that background fluorescence was negligible using a negative control sample (cells not expressing L1) that will have no signal using the chosen setting (*Figure 1—figure supplement 4*).

For quantification reported in *Figure 4A*, HeLa cells expressing ORFeus were plated on eight wells chamber, induced with doxycycline, fixed and stained as described in the method section. Images of cells stained with JH74 antibody against ORF1p and Alexa 647 conjugated secondary antibody (Invitrogen, prod. number A32733), were collected with the Arrayscan VTI and quantified using the Compartmental Analysis Bioapplication. Briefly, 16 fields per treatment were acquired at 20x magnification and $2 \times 2$ binning ($1104 \times 1104$ resolution). DAPI positive nuclei were identified using the dynamic isodata thresholding algorithm after minimal background subtraction. DAPI images were used to identify cell nuclei and to delineate the nuclear edges (x = 0). A 'circle' smaller than the nucleus (x=-4) was used to identify cells with nuclear ORF1p and a 'ring' outside the nucleus (1 < x > 8) was used to identify cells expressing cytoplasmic ORF1p. Limits of fluorescence were set so that no cells were considered positive for preparations of cells not treated with doxycycline (negative control). The reported parameters are explained below:

Total = total number of DAPI nuclei counted;

Nuclear ORF1p=cells with fluorescence signal inside the circle (x=-4) higher than the limit (higher fluorescence signal of the circle in the negative control)

Cytoplasmic/total ORF1p=cells with fluorescence signal inside the ring (1 < x > 8) higher than the limit (higher fluorescence signal of the ring in the negative control).

Because all cells expressing L1 show expression of ORF1p in the cytoplasm, the number of cells with cytoplasmic ORF1p was considered as the number of total cells expressing ORF1p. The percentage of cells with nuclear ORF1p was calculated as:

(number of cells with nuclear ORF1p)*100/(number of cells with cytoplasmic ORF1p=total number of cells expressing ORF1p).

All results are reported as mean and the error calculated as standard deviation (S.D.) or standard error of the mean (S.E.M). Data are considered to be statistically significant when p<0.05 by two-tailed Student's T test. In *Figure 5*, asterisks denote statistical significance as calculated by Student's T test (*p<0.05; **p<0.01; ***p<0.001).

## Proximity analysis

HeLa M2 cells expressing ORFeus for 24 hr were plated on fibronectin treated chamber-slides, treated with 0.1 µg/ml doxycycline, fixed and stained using JH74 rabbit primary antibody against ORF1p, Alexa 647 conjugated anti-rabbit secondary antibody and DAPI as described in the method section. Images were collected with the Arrayscan VTI system and cell positions within each slide obtained using the Compartmental Analysis Bioapplication. To quantify if cells with nuclear ORF1 are significantly closer to each other compared to random cells, we first calculated the shortest distance for each nuclear ORF1 cell to another nuclear ORF1 cell. Next, we randomly and repeatedly (n = 1000) select the same number of cells as there are nuclear ORF1 cells and obtain the distribution of distances that correspond to random localization of non-nuclear ORF1 cells. We used the random distribution to calculate the p-value and a false discovery rate (FDR). Finally, we compared the distribution of distances for nuclear ORF1 cells that are significantly closer to each other (p-value=0.1) with the distribution of a random sample that did not express nuclear ORF1 using a Wilcoxon rank sum test.

## Data and software availability

Mass spectrometry data for ORF2p and V5-PCNA sequential IP presented in *Figure 9* have been deposited in MassIVE archive under submission number MSV000081124 and in ProteomeXchange archive under submission number PXD006628.

## Acknowledgements

This work was supported by NIH grant P50GM107632 to JDB. The cytometry and cell sorting, High Throughput Biology (HTB) and Proteomic cores are partially supported by Laura and Isaac Perlmutter Cancer Center Support Grant, (NIH/NCI P30CA16087) and NYSTEM Contract C026719 (HTB core) and NIH/ORIP 1S10OD010582 grant (proteomic core). We thank Gregory Brittingham and Dr. Liam Holt for their help in collecting confocal images and movies and Dr. Timothee Lionnet for the help with RNA FISH. The model presented in *Figure 10* was constructed modifying available pictures from mindthegraph-Science Infographic Maker.

## Additional information

### Funding

| Funder | Grant reference number | Author |
| --- | --- | --- |
| National Institutes of Health | P50GM107632 | Jef D Boeke |
| National Cancer Institute | NIH/NCI P30CA16087 | Chi Y Yun |
| National Institutes of Health | 1S10OD010582 | Chi Y Yun |
| NYSTEM | Contract C026719 | Chi Y Yun |

The funders had no role in study design, data collection and interpretation, or the decision to submit the work for publication.

## Author contributions

Paolo Mita, Conceptualization, Resources, Data curation, Formal analysis, Supervision, Validation, Investigation, Visualization, Methodology, Writing—original draft, Project administration; Aleksandra Wudzinska, Resources, Data curation, Investigation, Methodology, Writing—review and editing; Xiaoji Sun, Data curation, Writing—review and editing; Joshua Andrade, Shruti Nayak, Resources, Data curation, Software, Formal analysis, Investigation, Methodology, Writing—review and editing; David J Kahler, Resources, Data curation, Software, Formal analysis, Investigation, Methodology; Sana Badri, Resources, Data curation, Software, Formal analysis, Validation, Investigation, Visualization, Methodology; John LaCava, Resources, Methodology, Writing—review and editing; Beatrix Ueberheide, Resources, Data curation, Software, Formal analysis, Supervision, Visualization, Methodology, Writing—review and editing; Chi Y Yun, Resources, Data curation, Software, Formal analysis, Supervision, Methodology; David Fenyö, Resources, Data curation, Software, Formal analysis, Supervision, Funding acquisition, Validation, Visualization, Methodology, Project administration, Writing—review and editing; Jef D Boeke, Conceptualization, Supervision, Funding acquisition, Project administration, Writing—review and editing

## Author ORCIDs

Paolo Mita (iD) http://orcid.org/0000-0002-2093-4906
John LaCava (iD) http://orcid.org/0000-0002-6307-7713
David Fenyö (iD) http://orcid.org/0000-0001-5049-3825
Jef D Boeke (iD) http://orcid.org/0000-0001-5322-4946

## Decision letter and Author response

Decision letter https://doi.org/10.7554/eLife.30058.040
Author response https://doi.org/10.7554/eLife.30058.041

# Additional files

## Supplementary files

• Supplementary file 1: RNA-FISH probe sequences. The DNA sequences of the RNA-FISH probes used to detect L1rp and ORFeus mRNA are reported.
DOI: https://doi.org/10.7554/eLife.30058.033

• Transparent reporting form
DOI: https://doi.org/10.7554/eLife.30058.034

## Major datasets

The following datasets were generated:

| Author(s) | Year | Dataset title | Dataset URL | Database, license, and accessibility information |
|---|---|---|---|---|
| Boeke JD | 2017 | LINE-1 and the cell cycle: protein localization and functional dynamics | https://massive.ucsd.edu/ProteoSAFe/dataset.jsp?task=e2ea6a4b189c45e5867-b097a19fd0d0f | Publicly available at the MassIVE repository (accession no. MSV0000 81124) |
| Boeke JD | 2017 | LINE-1 and the cell cycle: protein localization and functional dynamics | http://proteomecentral.proteomexchange.org/cgi/GetDataset?ID=PXD006628 | Publicly available at ProteomeXchange (accession no. PXD00 6628) |

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
