## [Decision Letter]

Thank you for submitting your article "LINE-1 and the cell cycle: protein localization and functional dynamics" for consideration by *eLife*. Your article has been reviewed by three peer reviewers, and the evaluation has been overseen by a Reviewing Editor and James Manley as the Senior Editor. The following individuals involved in review of your submission have agreed to reveal their identity: Suzanne Sandmeyer (Reviewer #2); Geoffrey Faulkner (Reviewer #3).

The reviewers have discussed the reviews with one another and the Reviewing Editor has drafted this decision to help you prepare a revised submission.

Summary:

The work presented in the paper addresses an issue in the retrotransposition field that is of long standing, of much interest to other areas (the cell cycle field), and of much significance in understanding aspects of trafficking and nucleic acid sensing. The major conclusion is that LINE1/L1 "hop" frequencies are affected by the cell cycle. Numerous readouts of L1 components are assayed and localized across the cell cycle, under various manipulations of the cycle.

Essential revisions:

The paper was uniformly viewed by the three reviewers as being of much interest and value. All three had very specific issues that they felt needed to be addressed/corrected before the paper would be suitable for acceptance. I include all their reviews with this response because I think their full comments will be valuable. With respect to their comments, the most important could be summarized here:

1) Reviewer 1 suggests adding more assays of the L1 nucleic acids, admitting that this is a major undertaking. I agree that this is probably too much to ask, though certainly any information the authors have available in this area would strengthen the paper.

The other issues here should be readily addressed, mostly by softening claims or interpretations.

Interpreting the cell cycle status seems to need revising.

The significance of the Orf2-containing complexes seems to need softening -- the assumptions about their position in the life cycle of retrotransposition need to be revised.

2) Reviewer 2:

The Orf1 localization, and again the significance and the basis of that localization, seem to be in question. Likely the conclusions here need to be softened, or at least better justified. An alternative explanation should be discussed, or the reasons for not doing so, should be offered.

The effects of synchrony on transposition have an alternative explanation that, again, should be addressed or dismissed, if there is a reason to do so.

The other requests for clarification should be addressed by revisions.

3) Reviewer 3:

All the requests for clarification here seem appropriate and can be addressed by revisions. Points 10 and 12 are more demanding. Point 10 in particular asks for more data, and I agree this would be attractive but may require much work. Likely this point can be handled by softening the interpretations, or offering alternatives. Point 12 is an issue of much interest to me, and I hope can be addressed in detail -- the timing of appearance of the readout is key, and needs to be brought up (probably right away in the description of the experiments).

We look forward to a revision addressing as many of the comments of the reviewers as possible.

Reviewer #1:

This manuscript was submitted as a companion to Malloy et al. although they are related and could possibly be merged into a single perhaps more robust report, I agree that as written these two co submitted datasets tell two separate but complementary stories that are strong enough to stand alone; this study provides exciting new information that addresses a controversy in the literature about whether LINE-1 retrotransposes in dividing or non-dividing cells and reports novel findings about the timing of different L1 intermediates across the cell cycle. The only major concern is that this work would be strengthened by co-identification/localization of L1 nucleic acid intermediates, ideally both RNA and cDNA, although this addition would be a major undertaking and should not be required before this substantial advance to the field can be shared. However, the lack of this information impacts and to some extent invalidates the conclusions as stated. My concerns are largely about the authors' interpretations and conclusions, and not the quality or robustness of the data which are excellent.

In descriptions of Figure 1, it appears that nuclear ORF2p has a punctate pattern in the top panel, but a uniform distribution in the bottom panel, or if punctate, co localizing with ORF1p which also appears faintly punctate in the bottom panel nuclear staining. Also, the data in Figure 1—figure supplement 1igure with JH73 seems to show more punctate nuclear ORF1p staining, as does Figure 2 and the first movie. Either better images are required or the text needs to back off from the claim that there is only uniform, non-punctate ORF1p nuclear staining and that this is completely distinct from the pattern of ORF2p staining.

In light of the accompanying manuscript by Malloy, the conclusion (subsection “Analysis and quantification of ORF1p and ORF2p expression and cellular Localization”) "impaired in binding to ORF2p….most of the nuclear ORF1p species fail to bind ORF2p" is somewhat misleading – most of the ORF2p detected overall is not bound to ORF1p in the absence of RNA and the RNA is a crucial transposition intermediate.

Some of the cell cycle interpretations are questionable because: Cyclebase3.0 indicates that Cdt1 is max at the G1/S boundary, not G1 (as repeatedly noted), and geminin increases at the beginning of S but is maximal beginning mid-S through midG2, then declines again. Also, the effects of leptomycin treatment are less than 2x. These kinetics and results call into question the interpretation in subsection “LINE-1 retrotransposition peaks during S phase” "Following early G1, ORF1p is then exported to the cytoplasm." Maybe instead TPRT occurs immediately prior to release of the cell cycle G1/S block (no reason to assume that block prevents ORF2 endo or RT function), ORF1p is displaced as RNA is copied to cDNA and then exported/degraded. This scenario is also consistent with the findings in Figure 4 with the nocodazole block.

Major concerns regarding the conclusions of subsection “ORF2p binds chromatin and localizes at replication forks with PCNA during S Phase” "Interestingly, as expected from our previous observations revealing that ORF1p is not necessarily in the complex(es) with chromatin bound retrotransposing L1RNPs, immunoprecipitation of ORF1p pulled down only a small amount of ORF2p, and also a smaller amount of MCM6 and PCNA proteins. These observations suggest that the retrotransposing L1 complex containing ORF2p, PCNA and components of the replication fork such as MCM6, are depleted of ORF1 proteins". What is the evidence that the ORF2p-PCNA complex is a "retrotransposing L1 complex"? Minimally it is necessary to show that L1 RNA is there since that is a key intermediate in the complex undergoing active retrotransposition. Perhaps the ORF2 that lingers on the DNA with PCNA is simply stable with an interference complex, or a repair complex after all of the key tprt steps (cleavage of target and reverse transcription of the RNA) are completed. What evidence eliminates either a pre-tprt non-productive interference complex or a post-tprt repair complex as an alternative explanation for the stable ORF2p nuclear complexes? The PCNA foci along with ORF2p could reflect the end repair required to complete insertion /heal the broken backbone presumably left by tprt-long after L1 RNA and (ORF1p) are gone. This concern also applies to the fifth paragraph of the Discussion section.

The fourth paragraph in the Discussion section suggests that there are at least two distinct complexes with ORF2p in the nucleus. "Conversely, not all ORF2p nuclear foci overlap with PCNA sites, suggesting that L1 interaction with the replication fork may represent just one of several modes used by L1 to select a DNA target site and retrotranspose." It is not clear why presume this (or "at least in some instances, retrotransposing L1 may specifically interact with a subset of perhaps stalled replication forks"), since it is a well-established feature of DNA repair that single strand breaks in DNA (as caused by tprt) stall replication forks. Thus, the observed association seems more likely to be a consequence of nearly completed tprt than "retrotransposing L1 specifically interacting with stalled forks during replication".

Reviewer #2:

L1 elements represent about 20% of the human genome and yet we still do not completely understand the formation of L1 RNPs and how the integration complex accesses the chromosomal DNA. The questions addressed in this ms are germane and the answers interesting. The Boeke laboratory is a leader in L1 biochemistry. Here they specifically address two key issues-the mechanism of localization of the L1 proteins encoded in the first and second open reading frames (ORFs) during the cell cycle in cells undergoing retrotransposition and the timing of retrotransposition itself. While the authors achieve a high level of reasonableness in their conclusions some rest on inferences from essentially negative results. Unfortunately, as authors note early in the manuscript, for technical reasons their conclusions may be most relevant to cells over-expressing L1 rather than cells in which native retrotransposition is occurring.

1) L1 encoded proteins bypass active nuclear import during M phase. ORF1 is enriched in the nucleus in cells expressing the Cdt1 G1 marker, but is weakly detected and diffuse in cells expressing the geminin S/G2/M marker. Based on this the authors argue that if ORF1 is high in G1 and lower in G2/S/M then it must enter the nucleus in M. However, it appears concentrated rather than simply partitioned. This suggests that ORF1 actively associates with chromatin, an idea that is actually supported by Figure 6 (IP). If ORf1 does not localize to nucleus by binding chromatin, authors should explain how M phase actually partitions ORF1 to the G1 nucleus over the cytoplasm. Given the reliance on IF, authors should include negative controls under experimental conditions for cells not expressing ORF1 and ORF2. These could be included in Supplemental.

Authors then investigate the decrease in nuclear ORF1 from G1 to G2/S/M. Initially they propose ORF1 could be degraded or exported. They inhibit CRM1 with leptomycin and see an increase in nuclear ORF1. They conclude that CRM1 export of ORF1 is responsible for the lower level of nuclear ORF1 in cells in S/G2/M compared to G1 (Figure 3). There are several problems with this conclusion: (1) there is no attempt to be quantitative, or show a NES, or a control exported protein, so this leaves room for protein turnover to explain the observations in Figure 3; (2) it is notable that authors ignore a possible converse mechanism for high levels of nuclear ORF1 in G1, ie. low CRM1 activity (this could be tested with leptomycin treatment of cells blocked in G1-S-if reduced CRM1 activity is not involved ORF1 by IF should remain the same); (3) finally, although IF suggests a drop in nuclear ORF1 with progression from G1, the IP in Figure 6 does not seem to support that as it shows nuclear ORF1 seeming to increase slightly or remain constant. Authors please comment.

2) Retrotransposition peaks in S. Here authors initially present three possibilities: retrotransposition could occur in M, S, or G2, but only directly arrest cells for testing in M phase. Authors please clarify interpretation of the M phase arrest experiment. If retrotransposition is induced prior to arrest and occurs in the course of their experiment (Figure 4AB), assuming no retrotransposition in M phase cells, there should be decreasing retrotransposition with increasing time of arrest across the histogram chart in 4B from 0 to 21 h. Instead, retrotransposition remains constant or increases slightly with longer nocodazole arrest.

Authors use doxy induction followed by nocodazole/shakeoff to synchronize cells to initiate the cell cycle in M phase and then test for correlation between cell cycle and retrotransposition (Figure 4). In this experiment the authors argue if there is no effect of cell cycle, then there will be a constant increase in retrotransposed cells and conversely if there is an effect, then retrotransposition will occur synchronously in the permissive phase of the cell cycle leading to a peak in the rate of accumulation of retrotransposed cells. However, there is no specific control for the effect of synchronous induction of L1 or the level of intermediates upon release from nocodazole arrest (ie. synchrony at the beginning could lead to synchrony at the time taken to complete retrotransposition). Implicit in authors' argument is that intermediates have accumulated starting with nocodazole arrest and are poised to integrate. It should be simple to show that RNA and proteins are accumulating from 0 to 12 (during M, G0, G1) in the Figure 4 experiment. One possibility is that they are at some equilibrium level obviating the synchronous induction problem.

The Figure 5 experiment is clever and convincing but conclusions should be more clearly stated. This experiment shows a peak of newly-expressed reporter (blue) in S, but it is difficult to ignore what appear to be significant shoulders in G2/M. Authors emphasize the S phase peak. For example, in the Discussion section they state ".it is during the following S phase in which retrotransposition peaks (Figure 4 and Figure 5). The finding that L1 retrotransposition has a strong bias for S phase is not surprising" and the Abstract concludes similarly that retrotransposition "peaks during S phase". Readers might take from this that retrotransposition is restricted to S when in fact authors are only concluding that it peaks in S. There are several reasons to be careful about leaving readers with this impression. These authors have previously shown that L1 mutants affected in RT activity do not co-IP with PCNA, so that association with targets could occur before PCNA is present (ie not necessarily at replication forks). Also, PCNA is known to associate with repair sites as well as replication forks, so that the association with targets might not require an active replication fork. Consistent with that, in NHEJ mutants there are differences in L1 integration, consistent with L1 integration possibly occurring in association with DNA repair, potentially in G2. I think Goodier et al. did not identify PCNA in their MS report on the L1 interactome. Please comment on experimental conditions if known that could explain why Goodier did not see PCNA (or correct this impression!), given the justifiable importance placed on PCNA.

Figure 6. Authors, please explain why nuclear ORF1 is not decreased in S and G2/M and why proteins are not increasing throughout since they are induced to be expressed and other proteins are presumably at steady state. Please comment on the significance of the fuzzy ORF1 band in nuclear soluble material. Is soluble ORF1 modified?

Figure 7. MS data are presented to support the interaction between ORF2 and PCNA and indirectly the argument for S phase. However, according to Materials and methods section, 279 proteins that were identified for query in the STRING database were searched " to identify known protein-protein interactions between these candidates". This sounds as though Figure 7 only shows proteins known to interact with PCNA. In addition, two peptides allowing one or two missed sites is stated as the threshold, but MS data could better be presented as a probability to account for differences in recovery of peptides based on the size of the protein as well as the efficiency of recovery of larger peptides. This presentation of the data seems to have been somewhat filtered to support the PCNA replication fork thesis. Authors should better justify the threshold for reporting and at least in Supplemental show a ranked list of the 279 proteins identified in the two step IP-MS.

Reviewer #3:

Mita et al. make some interesting observations regarding the dynamic nature of L1 retrotransposition during the cell cycle. Although these data are potentially of importance to the field, their presentation could be substantially improved and clarified, and it is necessary to avoid over-interpretation. Use of qualitative phrases like "ORF1p is always mostly cytoplasmic" and generalizations like "ORF1p is never nuclear in cells in S/G2/M phase"-the language is simultaneously too strong yet inexact. There were also some significant concerns relating to experimental design and interpretation that would need to be addressed prior to publication.

Specific points:

1) 17% of the human genome is made up of L1 copies-please cite Lander et al., 2001.

2) 5'UTR with bidirectional promoter-please cite Speek, 2001, Swergold, 1990

3) Open reading frames, 3'UTR, general L1 structure-please cite Dombroski, 1991

4) L1 cytoplasmic foci with stress granule proteins-please cite Doucet et al., 2010.

5) The assertion that L1 ORF1p-containing cytoplasmic foci are "presumed artefacts induced by non-physiological expression levels of ORF1p" is highly debatable. Martin and Branciforte, (1993) demonstrate the formation of endogenous ORF1p foci in mouse embryonal carcinoma cell lines.

6) Figure 1 immunofluorescence-quantitation?

7) Figure 2: Please provide a secondary antibody-only control to confirm that the ORF1p staining is specific and not due to background nonspecific staining. Supplemental images (Figure 1—figure supplement 2) and movies containing confocal microscopy and z-stack movies should be mentioned explicitly in the text. Confocal images should also be provided to confirm the results in Figure 2 and definitively show co-localization of L1 proteins with Ctd1 and Geminin.

8) Figure 2: no quantitation is provided for results from the FUCCI system.

9) Figure 3: What is on the y-axis of these graphs? Percentages seem unlikely when compared to Figure 3. Are they numerical cell counts? If so, how many total cells were analysed for each condition? How were the statistics done?

10) Figure 4: please include additional timepoints for nocodozole treatment lasting between 9 hours and 21 hours. There is a slight but non-significant increase in retrotransposition from 0-9 hours in nocodozole, and had cells been collected regularly after 9 hours the relative retrotransposition may have increased further. Compare Figure 4 to Figure 4. In Figure 4, the increase in retrotransposition during S-phase initiates between 10 and 15 hours after release from nocodozole. However, in Figure 4, cells are only held in nocodozole and measured regularly for up to 9 hours. Also in Figure 4 relevant control would be to assess retrotransposition along the same timecourse with cells that had not been treated with nocodozole, to demonstrate an appreciable increase in GFP+ cells in a normally cycling population.

11) Subsection “LINE-1 retrotransposition peaks during S phase”, the third paragraph begins describing the fluorescent timer reporter system, and there is a sentence midway through the paragraph which discusses GFP expression upon doxycycline induction and refers to Figure 5 and Figure 5—figure supplement 1. According to Figure 5—figure supplement 1 this is a control to demonstrate the kinetics of doxycycline induced gene expression, but it seems completely out of context. Please rewrite to make this paragraph more clear.

12) How long does it take for a newly-integrated engineered L1 insertion to result in detectable GFP (or FT) expression? If TPRT occurs during S-phase, would the cell actually "turn green" during S-phase? The kinetics of doxycycline-induced GFP expression in Figure 5—figure supplement 1 suggest that it takes around 3 hours from the time reporter gene expression is induced to achieve detectable expression in 50% of cells, and 8 hours to get detectable expression in 90% of cells. It seems that a similar delay would apply to expression of a reporter gene delivered by L1 retrotransposition.

---

## [Author Response]

Essential revisions:Reviewer #1:This manuscript was submitted as a companion to Malloy et al. although they are related and could possibly be merged into a single perhaps more robust report, I agree that as written these two co submitted datasets tell two separate but complementary stories that are strong enough to stand alone; this study provides exciting new information that addresses a controversy in the literature about whether LINE-1 retrotransposes in dividing or non-dividing cells and reports novel findings about the timing of different L1 intermediates across the cell cycle. The only major concern is that this work would be strengthened by co-identification/localization of L1 nucleic acid intermediates, ideally both RNA and cDNA, although this addition would be a major undertaking and should not be required before this substantial advance to the field can be shared. However, the lack of this information impacts and to some extent invalidates the conclusions as stated. My concerns are largely about the authors' interpretations and conclusions, and not the quality or robustness of the data which are excellent.

We thank the reviewer for the helpful and positive comments. In answer to the reviewer’s concerns we added data, performing L1 mRNA FISH and qPCR analysis following ORF2p and PCNA immunoprecipitations (Figure 3 and Figure 9). The new results fit and strengthen our previous conclusions and show high levels of nuclear L1 mRNA in cells exiting mitosis (Figure 3 and Figure 3—figure supplement 1 and Figure 3—figure supplement 2) and the binding of L1 mRNA to the ORF2p/PCNA complex (Figure 9). We also added all requested controls plus additional data using G1/S synchronization (Figure 5). Finally, we adjusted the main text to clarify and expand interpretations, especially concerning the role of the replication fork on the L1 life-cycle.

In descriptions of Figure 1, it appears that nuclear ORF2p has a punctate pattern in the top panel, but a uniform distribution in the bottom panel, or if punctate, co localizing with ORF1p which also appears faintly punctate in the bottom panel nuclear staining. Also, the data in Figure 1—figure supplement 1 with JH73 seems to show more punctate nuclear ORF1p staining, as does Figure 2 and the first movie. Either better images are required or the text needs to back off from the claim that there is only uniform, non-punctate ORF1p nuclear staining and that this is completely distinct from the pattern of ORF2p staining.

We agree to eliminate from the manuscript all comments regarding the differences between ORF1p and ORF2p punctate nuclear patterns as suggested.

In light of the accompanying manuscript by Malloy, the conclusion (subsection “Analysis and quantification of ORF1p and ORF2p expression and cellularLocalization”) "impaired in binding to ORF2p….most of the nuclear ORF1p species fail to bind ORF2p" is somewhat misleading – most of the ORF2p detected overall is not bound to ORF1p in the absence of RNA and the RNA is a crucial transposition intermediate.

In our previous paper (Taylor et al., 2013) we showed that ORF2p and ORF1p binding is RNA dependent (in vitrotreatment of L1 RNP with RNase abrogate co-immunoprecipitation of ORF1p and ORF2p); here (and in Molloy et al.) we show that a fraction of ORF2p does not colocalize/interact with ORF1p. Our IF staining and IPs with JH73g (that recognizes a nuclear form of ORF1p) and Molloy’s GO term analysis of ORF2p not bound to ORF1p (and most likely still bound to L1 mRNA) both support the idea that the ORF2p-only fraction is nuclear. These points are emphasized in the Discussion section.

Some of the cell cycle interpretations are questionable because: Cyclebase3.0 indicates that Cdt1 is max at the G1/S boundary, not G1 (as repeatedly noted), and geminin increases at the beginning of S but is maximal beginning mid-S through midG2, then declines again.

Many works have now used Cdt1 and geminin as markers of G1 and S/G2/M phase respectively (Aizer et al., 2013; Coleman et al., 2015; Sakaue-Sawano et al., 2008; Shiomi et al., 2014; Truong and Wu, 2011; Wohlschlegel et al., 2002; Zhao et al., 2014). Mechanistically, it is now known that Cdt1 is a key licensing factor that must be rapidly destroyed upon initiation of DNA replication. Geminin (GMNN) is a DNA Replication inhibitor that, together with absence of Cdt1, prevents spurious re-initiation of DNA replication during S/G2/M phase. Cdt1 and Geminin are highly regulated in the cell by post-translational modification and degradation by the Anaphase Promoting Complex (APC/C) in the case of Geminin and SCF-Skp2 complex in the case of Cdt1. Therefore, the critical factor is the protein (not mRNA) levels of Cdt1 and Geminin. We did check Cyclebase 3.0 for *CDT1* and *GMNN* (geminin) (we thank the reviewer for introducing us to this valuable resource) but found relevant data only on GMNN. Geminin transcripts, as mentioned by the reviewer, peak at early S, but the protein is expressed from late G1 to the end of mitosis rendering it a perfect marker for S/G2/M phases. The use of Geminin and Cdt1 as markers of S/G2/M and G1 are strongly corroborated by data on FUCCI cell lines (Bajar et al., 2016; Mechali and Lutzmann, 2008; Sakaue-Sawano et al., 2008; Sakaue-Sawano and Miyawaki, 2014). FUCCI cell lines use the “destruction” peptides from Cdt1 and Geminin conjugated to fluorophores for easy visualization of G1 and S/G2/M cell cycle stages respectively. We verified the expression of the proteins with live cell imaging and confirmed the previously published data. We are therefore confident that cells positively stained for Cdt1 are in G1 while cells positively stained for Geminin are in S/G2/M.

Also, the effects of leptomycin treatment are less than 2x. These kinetics and results call into question the interpretation on p7 "Following early G1, ORF1p is then exported to the cytoplasm."

We agree with the reviewer that the differences observed with leptomycin treatments are small but they are highly reproducible and have been repeated with several antibodies as presented in Figure 4. We attribute this absence of bigger effects to possible indirect roles of CRM1 in ORF1p nuclear export or the possible existence of multiple pathways for ORF1p export and/or degradation. Also, the technical difficulties of this type of experiment, i.e., treatment with low concentrations of leptomycin for a time allowing cells to go through at least one mitosis present a challenge. Treatments with higher concentrations of leptomycin induce inhibition of cell cycle progression and prevent cells from entering and exiting from mitosis that we know to be essential for entrance of L1 RNPs into the nucleus. We now present (Figure 4—figure supplement 2), controls (MEK1-GFP) showing the effectiveness of this leptomycin regimen, as also requested by reviewer #2. Finally, we stress another point strongly supporting our hypothesis that nuclear ORF1p cycles during the cell cycle. We have very clear results using FUCCI cell lines that show that no cells 4 in S phase present nuclear ORF1p staining (Figure 2 and subsection “Analysis and quantification of ORF1p and ORF2p expression and cellular localization”). This outcome would be impossible if ORF1p did not cycle into the cytoplasm (and/or get degraded in the nucleus) after G1 phase in populations of cycling cells induced to express L1 for more than 24 hrs. Indeed, if nuclear ORF1p is not degraded or exported back into the cytoplasm after one cell cycle (< 24h in HeLa cells) the protein localization of ORF1p would soon reach steady state and all cells would present nuclear ORF1p. This is clearly not the case (see Table in Figure 4).

Maybe instead TPRT occurs immediately prior to release of the cell cycle G1/S block (no reason to assume that block prevents ORF2 endo or RT function), ORF1p is displaced as RNA is copied to cDNA and then exported/degraded. This scenario is also consistent with the findings in Figure 4 with the nocodazole block.

To make sure that retrotransposition did not initiate in late G1 we performed similar experiments to the one presented in Figure 5 but instead of using nocodazole we treated cells with mimosine or thymidine to analyze the effect of arrest at the G1/S boundary. The results presented in Figure 5/D show no increase of retrotransposition with increasing time in late G1 supporting our idea that retrotransposition happens during S phase proper. This is also when PCNA joins the MCM complex to the replication fork. Interestingly, we observed a decrease in retrotransposition upon treatment with mimosine suggesting additional modes of L1 inhibition by this compound (see answer 7 to reviewer #2).

Moreover, we know from sequencing of L1 insertions into the genome, that the number of successful retrotranspositions per cell are few (one-two per cell over a period of days). Follow up experiments to the data presented here have led us to believe that most of the L1 recruited to DNA produces “aborted” retrotransposition due to action of homologous recombination repair machinery and/or other inhibitors. Recruitment of L1 RNP to DNA is a distinct step that is necessary but not sufficient for successful TPRT. We therefore argue that if ORF1p leaves the L1 RNP only during productive TPRT, undetectably few molecules would exit the nucleus or be degraded. We don’t think we could visualize that with standard fluorescence microscopy techniques such as those implemented here. Finally, if the reviewer’s hypothesis were true, in cells presenting high nuclear ORF2p and no nuclear ORF1p as for the cell shown in the top panel of Figure 1, all L1 RNPs should have gone through TPRT or at least one RT reaction to allow all ORF1p to be degraded or exported from the nucleus. This scenario seems unlikely given the overall inefficiency of retrotransposition. In our opinion, our results taken as a whole, are more easily explained by a general loss of ORF1p from nuclear L1 RNPs that is not directly linked to TPRT and probably precedes that step. This interpretation is strongly supported by the staining with JH73g antibody described in Figure 1—figure supplement 1. This antibody more specifically recognizes nuclear ORF1p that interacts less with ORF2p, supporting the existence of some still unknown difference between nuclear and cytoplasmic ORF1

Major concerns regarding the conclusions of subsection “ORF2p binds chromatin and localizes at replication forks with PCNA during SPhase” "Interestingly, as expected from our previous observations revealing that ORF1p is not necessarily in the complex(es) with chromatin bound retrotransposing L1RNPs, immunoprecipitation of ORF1p pulled down only a small amount of ORF2p, and also a smaller amount of MCM6 and PCNA proteins. These observations suggest that the retrotransposing L1 complex containing ORF2p, PCNA and components of the replication fork such as MCM6, are depleted of ORF1 proteins". What is the evidence that the ORF2p-PCNA complex is a "retrotransposing L1 complex"? Minimally it is necessary to show that L1 RNA is there since that is a key intermediate in the complex undergoing active retrotransposition.

The reviewer correctly points out that we lack direct data to unequivocally confirm that the PCNA-ORF2p complex is the L1 retrotransposing complex. We perhaps prematurely “labeled” this complex as the retrotransposing complex for the following reasons:

- ORF2p bound to PCNA is found on chromatin (Figure 7 and unpublished data from our laboratory aimed to identify ORF2p interacting proteins from cellular chromatin fractions using mass spectrometry analysis)

- ORF2p with mutated PIP domain is retrotransposition incompetent (Taylor et al., 2013)

- L1 RNPs isolated by ORF2p IP co-immunoprecipitating PCNA, are competent for in vitro LEAP and RT activity (Taylor et al., 2013)

- As our new results show, the PCNA-ORF2p complex also contains L1 mRNA (Figure 9)

Unfortunately, to our knowledge, there is no published *in-vivo* or *in-vitro* method to demonstrate that the PCNA-ORF2p complex is the actual complex involved in TPRT. Our laboratory is heavily invested in trying to develop an in vitro TPRT assay that would measure the ability of a specific ORF2p complex to cut a target DNA, retro-transcribe L1 mRNA, and reinsert its cDNA into the target DNA. Despite some advances toward this goal we are currently unable to definitively and directly assay the “TPRT capability” of a specific protein complex in vitro or in vivo. Based on the available data previously published and presented in this manuscript and in the Molloy et al. co-submitted manuscript, the most likely complex to be a retrotransposing complex is one comprising ORF2p, L1 RNA and PCNA. We agree with the reviewer that, technically, defining the PCNA-ORF2p complex as the “L1 retrotransposing complex” is premature and therefore we changed our wording to “retrotransposition-competent complex” in the Discussion section and to “nuclear complex” in subsection “ORF2p binds chromatin and localizes at replication forks with PCNA during S phase” and “L1 possibly engaged in TPRT” in the Discussion section.

Perhaps the ORF2 that lingers on the DNA with PCNA is simply stable with an interference complex, or a repair complex after all of the key tprt steps (cleavage of target and reverse transcription of the RNA) are completed. What evidence eliminates either a pre-tprt non-productive interference complex or a post-tprt repair complex as an alternative explanation for the stable ORF2p nuclear complexes? The PCNA foci along with ORF2p could reflect the end repair required to complete insertion /heal the broken backbone presumably left by tprt-long after L1 RNA and (ORF1p) are gone. This concern also applies to the fifth paragraph of the Discussion section.

The reviewer asks an excellent question that we would also re-phrase as: “is ORF2p recruited to chromatin by PCNA or is PCNA recruited to the site of retrotransposition by ORF2p?”. In our previous work (Taylor et al., 2013) we showed that ORF2p EN or RT mutants do not interact directly with PCNA. Our unpublished data also show that the ORF2p PIP mutant, incapable of interacting with PCNA (Taylor et al., 2013) is still able to bind chromatin. These data suggest that retrotransposing ORF2p recruits PCNA to the 6 DNA. Then how do we justify ORF2p binding to the MCM complex? It is possible that ORF2p is recruited to the replication fork through another mechanism, and that after the initial steps of TPRT, interaction with PCNA, readily available at the replication fork, becomes essential for the completion/repair of the L1 reinsertion in the genome. Supporting this idea, we find ligase 1/LIG1 (that interacts with PCNA on replication fork and mediates ligation of the Okazaki fragments on the lagging strand) as an ORF2p interactor in chromatin bound fractions from cycling HeLa cells (unpublished data). We therefore hypothesize that PCNA is recruited to ORF2p after the endonuclease cleavage and right after RT engagement with L1 mRNA template but before completion of TPRT and final ligation of L1 cDNA into the genome. This would justify the fact that many molecules of PCNA strongly interact with ORF2p as judged by co-IP experiments (Figure 9 and Figure 8—figure supplement 1).

Throughout the manuscript we never mentioned whether ORF2p is recruited by PCNA onto chromatin (or vice versa) because the model explained above is, in our opinion, still too speculative. We realize though that this approach may mislead the readers and therefore, to incorporate these more speculative aspects of our model, we added a paragraph to the Discussion section about this subject. We also modified our graphical abstract eliminating the arrow depicting retrotransposition that might mislead the readers into thinking that retrotransposition happened via PCNA-mediated recruitment of L1 onto host DNA.

The fourth paragraph in the Discussion section suggests that there are at least two distinct complexes with ORF2p in the nucleus. "Conversely, not all ORF2p nuclear foci overlap with PCNA sites, suggesting that L1 interaction with the replication fork may represent just one of several modes used by L1 to select a DNA target site and retrotranspose." It is not clear why presume this (or "at least in some instances, retrotransposing L1 may specifically interact with a subset of perhaps stalled replication forks"), since it is a well-established feature of DNA repair that single strand breaks in DNA (as caused by tprt) stall replication forks. Thus, the observed association seems more likely to be a consequence of nearly completed tprt than "retrotransposing L1 specifically interacting with stalled forks during replication".

The reviewer raised, again, a very interesting point that we are still investigating in our follow up work to this manuscript. Is ORF2p recruited specifically to stalled replication forks or is it the DNA nick made by ORF2p endonuclease that stalls the replication fork? In our present model both possibilities are plausible. The sentences in the manuscript cited by the reviewer only aim to explain why not all ORF2p chromatin foci observed in Figure 9 coincide with a PCNA focus. We have softened and clarified the concepts in the text (Discussion section).

We lean towards a model in which L1 RNPs are recruited to stalled replication forks, and not vice-versa, because of an experiment we performed treating cells with low concentrations of aphydicolin, a compound that induces stalling of replication forks. Upon treatment, retrotransposition is increased suggesting that the stalling caused retrotransposition and not that retrotransposition caused the stalling. One interpretation of these experiment is that L1 RNPs, dragged along with the replication fork, may use 7 stalling as a signal to start retrotransposition. This hypothesis would also explain the strong interaction between ORF2p, PCNA and MCM proteins reported here. In our opinion, collision of the replication fork to retrotransposing ORF2p seems less likely to survive two sequential IPs as presented in Figure 9.

The question, however, remains still open to future investigations.

We added more details about our model and about L1 chromatin recruitment in the Discussion section.

Reviewer #2:L1 elements represent about 20% of the human genome and yet we still do not completely understand the formation of L1 RNPs and how the integration complex accesses the chromosomal DNA. The questions addressed in this ms are germane and the answers interesting. The Boeke laboratory is a leader in L1 biochemistry. Here they specifically address two key issues-the mechanism of localization of the L1 proteins encoded in the first and second open reading frames (ORFs) during the cell cycle in cells undergoing retrotransposition and the timing of retrotransposition itself. While the authors achieve a high level of reasonableness in their conclusions some rest on inferences from essentially negative results. Unfortunately, as authors note early in the manuscript, for technical reasons their conclusions may be most relevant to cells over-expressing L1 rather than cells in which native retrotransposition is occurring.

We thank the reviewer for the insightful comments and for acknowledging the contribution of our manuscript. We gave extensive explanations about the logic that lead to our conclusions that we hope will convince the reviewer of the validity of our model. We also integrated several paragraphs in the main text to clarify our views as requested.

1) L1 encoded proteins bypass active nuclear import during M phase. ORF1 is enriched in the nucleus in cells expressing the Cdt1 G1 marker, but is weakly detected and diffuse in cells expressing the geminin S/G2/M marker. Based on this the authors argue that if ORF1 is high in G1 and lower in G2/S/M then it must enter the nucleus in M. However, it appears concentrated rather than simply partitioned. This suggests that ORF1 actively associates with chromatin, an idea that is actually supported by Figure 6 (IP). If ORf1 does not localize to nucleus by binding chromatin, authors should explain how M phase actually partitions ORF1 to the G1 nucleus over the cytoplasm.

The reviewer brings up a very interesting point that we partially explored. ORF1p is a highly positively charged protein (unsurprising considering it is an RNA binding protein) and it likely engages in at least weak interaction with chromatin during M phase when ORF1p has access to DNA. Interestingly, a low percentage of L1 expressing cells transitioning through M phase shows a strong chromatin recruitment of ORF1p during metaphase. A distortion caused by a C-terminal GFP-tag on ORF1p strongly enhances such chromatin recruitment (unpublished data). Moreover, as the reviewer already noticed, we always observe chromatin recruitment of ORF1p upon cellular fractionation (Figure 9 and Figure 8—figure supplement 1). We can see recruitment of ORF1p on chromatin even in the absence of ORF2p (expressing constructs lacking ORF2p) suggesting that ORF1p chromatin recruitment is independent of ORF2p and retrotransposition (unpublished data). We therefore lean towards the possibility that ORF1p has weak interactions with chromatin that facilitate the localization of L1 RNPs onto chromatin and the post-mitotic nuclear localization of L1. But the nuclear import of proteins lacking a nuclear localization signal is still poorly understood. It is accepted that proteins can be “trapped” in the nucleus during mitosis and after nuclear membrane reformation.

We added a paragraph about these interesting hypotheses in the Discussion section.

Given the reliance on IF, authors should include negative controls under experimental conditions for cells not expressing ORF1 and ORF2. These could be included in Supplemental.

We added a new Supplementary figure to Figure 1 (Figure 1—figure supplement 4) presenting the negative controls for our IF staining: secondary only controls and staining of non-expressing cells.

Authors then investigate the decrease in nuclear ORF1 from G1 to G2/S/M. Initially they propose ORF1 could be degraded or exported. They inhibit CRM1 with leptomycin and see an increase in nuclear ORF1. They conclude that CRM1 export of ORF1 is responsible for the lower level of nuclear ORF1 in cells in S/G2/M compared to G1 (Figure 3). There are several problems with this conclusion: (1) there is no attempt to be quantitative, or show a NES, or a control exported protein, so this leaves room for protein turnover to explain the observations in Figure 3;

As requested by the reviewer we added a Supplemental figure (Figure 4—figure supplement 2) showing nuclear retention of a known CRM1 regulated protein (MEK1-GFP) in HeLa-M2 cells treated with leptomycin for different amounts of time. This serves as control showing that treatment of cells with leptomycin induces nuclear retention of CRM1 regulated proteins.

(2) it is notable that authors ignore a possible converse mechanism for high levels of nuclear ORF1 in G1, ie. low CRM1 activity (this could be tested with leptomycin treatment of cells blocked in G1-S-if reduced CRM1 activity is not involved ORF1 by IF should remain the same);

The reviewer suggests that CRM1 levels and activity during the cell cycle may be responsible for the increased ORF1p we observed during early G1. We are not aware of CRM1 activity or cellular expression fluctuation during the cell cycle. Cyclebase 3.0 (Santos et al., 2015) reports CRM1/XPO1 mRNA expression peaking at the boundary of M / G1 and decreasing in late G1 /early S (a behavior opposite of what one would imagine if CRM1 level was regulating nuclear export of ORF1p in late G1/S). Also, protein expression of CRM1 is reported as high and constant during the whole cell cycles showing no fluctuation, suggesting that CRM1 cellular expression is probably not involved in ORF1p (and other proteins) nuclear localization.

Moreover, we are quite confident that the increased ORF1p in early G1 results from L1 RNPs entering the nucleus during mitosis. Several evidence types support this:

- cells with high nuclear ORF1p staining are closer together compared to cells with cytoplasmic ORF1p suggesting that these cells recently exited mitosis (Figure 1).

- our unpublished data on live cell imaging tracking ORF1p tagged with GFP clearly showed that ORF1p can get “trapped” in the nucleus upon transition through mitosis. Unfortunately, GFP seems to exacerbate ORF1p chromatin binding (see answer to question 1) rendering these results, in our opinion, less convincing than the ones presented in the manuscript.

(3) finally, although IF suggests a drop in nuclear ORF1 with progression from G1, the IP in Figure 6 does not seem to support that as it shows nuclear ORF1 seeming to increase slightly or remain constant. Authors please comment.

Please see response to question 9 of this reviewer.

2) Retrotransposition peaks in S. Here authors initially present three possibilities: retrotransposition could occur in M, S, or G2, but only directly arrest cells for testing in M phase. Authors please clarify interpretation of the M phase arrest experiment. If retrotransposition is induced prior to arrest and occurs in the course of their experiment (Figure 4AB), assuming no retrotransposition in M phase cells, there should be decreasing retrotransposition with increasing time of arrest across the histogram chart in 4B from 0 to 21 h. Instead, retrotransposition remains constant or increases slightly with longer nocodazole arrest.

The cells treated with nocodazole are initially asynchronous (usually starting at about 40-50% G1, 20-30% S and 20-30% G2/M). Upon nocodazole treatment, cells in mitosis (about 10%) arrest while the rest proceed through the cell cycle to stop at mitosis. Mitotic cells are then collected by mechanical shake off (this point is now clarified in the Materials and methods section and the legend of Figure 5) and analyzed for% of GFP+ cells. In the 21 hrs considered for this experiment there is no time for a cell to go through more than one cell cycle (and thus through >1 S phase) even in the case of no nocodazole treatment. This means that the presented experiment is assessing only retrotransposition in cells that spent increasing times in M phase upon nocodazole treatment going through just one S phase. If the experiments could be extended for more than 21 hrs (already a 11 time point showing increased cell death) retrotransposition would decrease compared to the control cycling population.

We also added to the revised manuscript similar analysis for cells synchronized in G1 phase by treatment with mimosine 1mM and thymidine 4mM (Figure 5).

Authors use doxy induction followed by nocodazole/shakeoff to synchronize cells to initiate the cell cycle in M phase and then test for correlation between cell cycle and retrotransposition (Figure 4). In this experiment the authors argue if there is no effect of cell cycle, then there will be a constant increase in retrotransposed cells and conversely if there is an effect, then retrotransposition will occur synchronously in the permissive phase of the cell cycle leading to a peak in the rate of accumulation of retrotransposed cells. However, there is no specific control for the effect of synchronous induction of L1 or the level of intermediates upon release from nocodazole arrest (ie. synchrony at the beginning could lead to synchrony at the time taken to complete retrotransposition). Implicit in authors' argument is that intermediates have accumulated starting with nocodazole arrest and are poised to integrate. It should be simple to show that RNA and proteins are accumulating from 0 to 12 (during M, G0, G1) in the Figure 4 experiment. One possibility is that they are at some equilibrium level obviating the synchronous induction problem.

The reviewer has concerns about possible effects of synchronization per se on retrotransposition. This was our concern when we designed experiments presented in Figure 7 that analyzes cycling pools of cells without needing synchronization. The results obtained with the use of fluorescent timer-AI constructs, mirrored the results obtained synchronizing the cells with nocodazole so we deemed these latter experiments worthy of publication. A (less elegant) alternative would have been the use of other synchronization methods. Unfortunately, the only other synchronization methods that we are aware of are not as effective as nocodazole, allowing many cells to escape the block. Other approaches require sequential synchronizations and are not easily adapted to our retrotransposition assay.

Finally, we have no reason to believe that the expression of L1 (ORF1p) under the Tet-promoter is regulated by the cell cycle. We also show in Figure 2—figure supplement 2 that ORF2p is expressed equally in G1 or S/G2/M cells. We therefore exclude a cell cycle effect on ORF2p translation.

The Figure 5 experiment is clever and convincing but conclusions should be more clearly stated. This experiment shows a peak of newly-expressed reporter (blue) in S, but it is difficult to ignore what appear to be significant shoulders in G2/M.

In Figure 5 (now Figure 7) we presented two experiments using cells expressing an ORFeus L1 with a fluorescent timer-AI cassette in its 3’UTR. Cells were treated for 24 h with doxycycline to induce expression of L1. In panel B, blue only cells (blue shaded histogram) and unstained cells from the same population (black and white histogram) were sorted and directly fixed in ethanol. The cells were then post-stained with propidium iodide to analyze the cell cycle stage of these cells. The result showed that blue-only cells (cells that completed retrotransposed within 3 hrs preceding sorting) accumulate mainly in S phase. In this setting we also observed a big shoulder of cells in G2/M phase as 12 pointed out by the reviewer. Therefore, we decided to use an experimental setting that does not require fixation and sorting but relies on the direct measurement of the cell cycle stage of different populations: non-fluorescent cells (black and white line), blue-only cells (blue shaded histogram) and blue cells with a very low red fluorescence (red shaded histogram). Using SYTO61 dye, that does not interfere with the fluorescent timer fluorescence, for the analysis of the cell cycle allowed us to easily gate the different cell populations and get a deeper insight into the big peak observed in panel A. We found that the peak shown in panel B actually consists of two sub-peaks: a peak of cells that are pure blue and in S phase and a peak of cells that progressed further into the cell cycle after retrotransposition and therefore display a blue/low red signal usually hidden by the background in analysis like the one shown in panel B. We therefore confirmed that retrotransposing cells are mainly in S phase.

To help clarify the logic behind the experimental design of Figure 7/B) we added a few lines to the description of the figure in the Results section.

Authors emphasize the S phase peak. For example, in the Discussion section they state ".it is during the following S phase in which retrotransposition peaks (Figure 4 and Figure 5). The finding that L1 retrotransposition has a strong bias for S phase is not surprising" and the Abstract concludes similarly that retrotransposition "peaks during S phase". Readers might take from this that retrotransposition is restricted to S when in fact authors are only concluding that it peaks in S. There are several reasons to be careful about leaving readers with this impression. These authors have previously shown that L1 mutants affected in RT activity do not co-IP with PCNA, so that association with targets could occur before PCNA is present (ie not necessarily at replication forks). Also, PCNA is known to associate with repair sites as well as replication forks, so that the association with targets might not require an active replication fork. Consistent with that, in NHEJ mutants there are differences in L1 integration, consistent with L1 integration possibly occurring in association with DNA repair, potentially in G2

Like reviewer #1, this reviewer raises a very interesting point: is L1 recruited on chromatin by PCNA or, vice-versa, PCNA is recruited by L1 on the site of retrotransposition. We think that PCNA is recruited to the L1 complex bound to chromatin in late stages of TPRT after genomic DNA cut and initiation of L1 RT (please see response 7 to reviewer 1).

We do not think that the MCM proteins are found in complex with ORF2p and PCNA simply because of collision of the replication fork with the site of DNA damage created by L1 as already discussed in the responses to reviewer 1 (please see response 8 to reviewer 1). As mentioned in our previous response to reviewer 1 we added a paragraph to the Discussion section about this subject and also modified our graphical model (Figure 10)

I think Goodier et al. did not identify PCNA in their MS report on the L1 interactome. Please comment on experimental conditions if known that could explain why Goodier did not see PCNA (or correct this impression!), given the justifiable importance placed on PCNA.

Goodier et al., (2013) performed a quite extensive analysis of L1 ORF1p interacting proteins but did not examine the ORF2p interactome. Our data show that nuclear and chromatin bound ORF2p is in a complex with PCNA and MCM proteins and depleted of ORF1p.

Figure 6 authors, please explain why nuclear ORF1 is not decreased in S and G2/M and why proteins are not increasing throughout since they are induced to be expressed and other proteins are presumably at steady state. Please comment on the significance of the fuzzy ORF1 band in nuclear soluble material. Is soluble ORF1 modified?

Nuclear fractionations (nuclei isolated and pelleted from low detergent or hypotonic buffer treated cells) are not as clean as chromatin fractions (proteins released from chromatin after micrococcal nuclease treatment). We therefore do not think that conventional fractionations can efficiently separate nuclear ORF1p from cytoplasmic ORF1p. We observed pelleting of cytoplasmic stress granules containing large amounts of ORF1p during nuclear fractionation, as shown by the G3BP1 marker in the chromatin fraction presented in Figure 6 (now Figure 8—figure supplement 1). In Figure 6 the nuclear fraction was presented for comparison with the chromatin fraction used for analysis. This is probably the reason why we could not observe differences in nuclear and cytoplasmic amount of ORF1p during the cell cycle in Figure 6 (now Figure 8). A short explanation has now been added to the main text (subsection “LINE-1 retrotransposition peaks during S phase”).

The presented blots show the expression of ORF2p, ORF1p and other proteins in two cellular fractions (chromatin and nuclear fraction) throughout one cell cycle. Other fractions have not been analyzed (cytoplasmic, membrane and “pellet” fraction). To analyze changes in overall expression of ORF1p and ORF2p all of the fractions should be summed or a total lysed fraction needs to be considered. Moreover, at time zero the cells have been already treated with doxycycline for 19 hours. In our experience, upon doxycycline treatment, the overall expression of L1 proteins reaches steady state after 20-24 hrs and therefore we don’t think that differences in total expression would be observed in our experiment.

The apparent fuzziness of ORF1p in nuclear fractions comes from the fact that we presented a “low exposure” blot because of the very high expression of ORF1p in the nuclear fraction compared to chromatin fraction. We preferred to present a “lower exposure” to better show possible differences in ORF1p nuclear fractions throughout the cell cycle. We clarify also that we use measurement of direct band fluorescence with LiCor CLx to quantify our blots. This analysis is completely independent of “exposure” and therefore, how to present the considered blot, is simply a “stylistic decision”.

Figure 7. MS data are presented to support the interaction between ORF2 and PCNA and indirectly the argument for S phase. However, according to Materials and methods section, 279 proteins that were identified for query in the STRING database were searched " to identify known protein-protein interactions between these candidates". This sounds as though Figure 7 only shows proteins known to interact with PCNA. In addition, two peptides allowing one or two missed sites is stated as the threshold, but MS data could better be presented as a probability to account for differences in recovery of peptides based on the size of the protein as well as the efficiency of recovery of larger peptides. This presentation of the data seems to have been somewhat filtered to support the PCNA replication fork thesis. Authors should better justify the threshold for reporting and at least in Supplemental show a ranked list of the 279 proteins identified in the two step IP-MS.

We previously demonstrated the direct interaction of ORF2p with PCNA through the PIP domain of ORF2p (Taylor et al., 2013). The Mass spectrometry analysis presented here was performed to identify additional proteins interacting specifically with the ORF2p/PCNA complex. Our main objective in performing this analysis was to better understand the role of PCNA in the L1 RNPs. As the reviewer mentioned, PCNA plays an important role in both DNA repair and DNA replication so we wanted to understand whether the ORF2p/PCNA complex had anything to do with DNA damage, DNA replication, both or neither. Therefore, we specifically looked in the obtained lists of ORF2p/PCNA interactors for proteins known to interact with PCNA leading us to a specific localization of the L1 RNP on chromatin. It was pretty clear to us, from the identification of the proteins reported in reviewed Figure 9 and from the validations in Figure 9, that other replication fork proteins are associated, suggesting that the fork itself plays a role in the L1 life-cycle. We therefore validated the interaction of MCM6 to L1 proteins by conventional IP-Western analysis (Figure 9). Our finding, of L1 interacting with the replication fork fits well with data showing retrotransposition occurs mainly during S.

The peptide and protein IDs were filtered using a 1% peptide and protein False discovery rate (FDR) when searched against a decoy database as specified in the Materials and methods section. In addition, we added an extra filter and required at least 2 unique peptides per protein before we considered it ID’d. This is an established standard in the mass spectrometry field. Likewise, using spectral counts as a measure of the relative abundance of a protein in 2 different samples is common practice in the field as accepted for comparative proteomics.

We agree with the reviewer that it would be beneficial to show all the data and therefore attached a new supplementary file (Supplementary file 2) that shows the result output for all 4 affinity purifications. The entire dataset is also available at Massive and ProteomXchange under accession numbers MSV000081124 and PXD006628 respectively.

We also added the number of identified proteins in the main text (subsection “LINE-1 retrotransposition peaks during S phase”) to avoid misleading readers into thinking that the proteins presented in Figure 9 were the only ones identified.

We realized that the numbers reported in the “mass spectrometry analysis” material and methods were referring just to one of the two MS analyses. We corrected this oversight.

Reviewer #3:Mita et al. make some interesting observations regarding the dynamic nature of L1 retrotransposition during the cell cycle. Although these data are potentially of importance to the field, their presentation could be substantially improved and clarified, and it is necessary to avoid over-interpretation. Use of qualitative phrases like "ORF1p is always mostly cytoplasmic" and generalizations like "ORF1p is never nuclear in cells in S/G2/M phase"-the language is simultaneously too strong yet inexact. There were also some significant concerns relating to experimental design and interpretation that would need to be addressed prior to publication.

We thank the reviewer for the deep analysis of our results and the help in making our manuscript more thorough and clear. We added new images and clarifications that show all requested controls and we hope allay reviewer concerns.

Specific points:1) 17% of the human genome is made up of L1 copies-please cite Lander et al., 2001.

We added the suggested reference.

2) 5'UTR with bidirectional promoter-please cite Speek, 2001, Swergold, 1990

We added the suggested reference.

3) Open reading frames, 3'UTR, general L1 structure-please cite Dombroski, 1991

We added the suggested reference.

4) L1 cytoplasmic foci with stress granule proteins-please cite Doucet et al., 2010.

We added the suggested reference.

5) The assertion that L1 ORF1p-containing cytoplasmic foci are "presumed artefacts induced by non-physiological expression levels of ORF1p" is highly debatable. Martin and Branciforte, (1993) demonstrate the formation of endogenous ORF1p foci in mouse embryonal carcinoma cell lines.

We softened the sentence in which we refer to L1 cytoplasmic foci and added references to paper supporting their physiological role in L1-life cycle.

6) Figure 1 immunofluorescence-quantitation?

Figure 1 shows a subset of cells quantified in Figure 1 which show representative fields. Figure 1 was meant to show the range of ORF1p and ORF2p localizations (both cytoplasmic, ORF2p nuclear ORF1p cytoplasmic etc.) in cells. We think the presentation of both Figure 1 (and its quantification) and Figure 1 is a more “honest” approach, instead of presenting just pictures in line with our conclusions.

7) Figure 2: Please provide a secondary antibody-only control to confirm that the ORF1p staining is specific and not due to background nonspecific staining. Supplemental images (Figure 1—figure supplement 2) and movies containing confocal microscopy and z-stack movies should be mentioned explicitly in the text. Confocal images should also be provided to confirm the results in Figure 2 and definitively show co-localization of L1 proteins with Ctd1 and Geminin.

We added a new Supplementary figure with all the corresponding negative controls for the presented IF staining (Figure 1—figure supplement 4). We clearly mention the movies and confocal images in the main text as suggested by the reviewer. We added a new supplementary figure (Figure 2—figure supplement 1) to include confocal images of Figure 2 as requested by the reviewer.

8) Figure 2: no quantitation is provided for results from the FUCCI system.

We did not provide quantification of FUCCI cell staining because, of all cells counted (3309), no cells with green nuclei (cells in S/G2/M) showed nuclear ORF1p. We were on the other hand able to find cells (80) with clear nuclear staining and red nuclei (in G1). The strong sentence “ORF1p is never (0/3309 cells counted) nuclear in cells in S/G2/M phase" is therefore just the product of a “strong” /very clear result. We added the numbers to the main text to make the description more quantitative (subsection “Analysis and quantification of ORF1p and ORF2p expression and cellularlocalization”).

9) Figure 3: What is on the y-axis of these graphs? Percentages seem unlikely when compared to Figure 3. Are they numerical cell counts? If so, how many total cells were analysed for each condition? How were the statistics done?

The label of the y-axis in the graphs in Figure 3 (now Figure 4) has been added (% of cells with nuclear ORF1p). As mentioned in subsection “ORF1p enters the nucleus during mitosis” and in subsection“Quantification of ORF1p and ORF2p cellular localization”, Figure 3 (now Figure 4) data were automatically acquired and analyzed by Arrayscan and Image Studio HCS software. This approach eliminates user biases but needs more stringent thresholds for specifying nuclear ORF1p because of the bright and high expression of ORF1p in the cytoplasm that easily interferes with automated analysis of the pictures.

As for all quantifications of L1 protein presented in the manuscript, the statistics were done using two tailed T-test on at least 1000 cells counted from each of at least 3 different preparations. As reported in the material and method section: “Quantification of cellular localization of ORF1p and ORF2p” was performed counting more than 1000 cells in three different cell preparations. For each experiment, LED intensity, camera gain and exposure were kept constant and set so that background fluorescence was negligible using a negative control sample (cells not expressing L1) that will have no signal using the chosen setting”.

10) Figure 4: please include additional timepoints for nocodozole treatment lasting between 9 hours and 21 hours. There is a slight but non-significant increase in retrotransposition from 0-9 hours in nocodozole, and had cells been collected regularly after 9 hours the relative retrotransposition may have increased further. Compare Figure 4 to Figure 4. In Figure 4, the increase in retrotransposition during S-phase initiates between 10 and 15 hours after release from nocodozole. However, in Figure 4, cells are only held in nocodozole and measured regularly for up to 9 hours.

Comparisons between Figure 4 (now Figure 5) cannot be made because the experiments in B were done treating cells for increasing times of nocodazole while experiment E shows retrotransposition after release into the cell cycle upon synchronization for 12 hrs with nocodazole. In Figure 4 (now Figure 5) there are actually no significant differences. We expanded the figure in any case (now Figure 5) to incorporate treatments with thymidine and mimosine that block cells in late G1/early S phase before DNA replication.

Also in Figure 4 relevant control would be to assess retrotransposition along the same timecourse with cells that had not been treated with nocodozole, to demonstrate an appreciable increase in GFP+ cells in a normally cycling population.

As shown in the schematic in Figure 4 (now Figure 6) the experiment is set up to specifically ask if increased times in treatments with a synchronization agent (nocodazole, mimosine or thymidine as in the revised manuscript) can affect retrotransposition compared to untreated cells (please see also response 5, reviewer #2). The samples are therefore all dox-treated, collected and analyzed together at the end of the experiment. Consequently, having an untreated well for each time point would have resulted in having 6 different replicate wells untreated with synchronization agent at the moment of analysis. The control the reviewer is asking is actually presented in Figure 4 (now Figure 6) in which the increase of retrotransposition of cells not synchronized with nocodazole (or any other agents) is shown.

11) Subsection “LINE-1 retrotransposition peaks during S phase”, the third paragraph begins describing the fluorescent timer reporter system, and there is a sentence midway through the paragraph which discusses GFP expression upon doxycycline induction and refers to Figure 5 and Figure 5—figure supplement 1. According to Figure 5—figure supplement 1 this is a control to demonstrate the kinetics of doxycycline induced gene expression, but it seems completely out of context. Please rewrite to make this paragraph more clear.

Quantification of the dynamics of GFP expression presented in Figure 5—figure supplement 1 (now Figure 7—figure supplement 1) aims to provide insight into the time needed for a fluorescent protein to be transcribed, translated and detected. We wanted to show that, given a successful retrotransposition, transcription and translation of a fluorescent reporter can occur within less than 3 h, maning retrotransposition was completed within the past 3 hours. With the presented measurements we showed that the expression from the CMV promoter is fast enough to allow detection of FT proteins still emitting blue light because they did not yet mature into red emitting proteins. In other experiments we used a weaker promoter (UBC) to drive the expression of the FT-AI in the L1-FT-AI constructs. Using these constructs, we were unable to identify blue cells. Only red cells were found in the L1 expressing and retrotransposing cell population, suggesting that the time necessary for the retrotransposed FT-AI gene to be transcribed, translated and accumulate to a measurable amount was longer than 3 h (the time of 19 maturation of the FT proteins from blue to red). These experiments also suggest that the strength of the promoter used to drive the AI reporter is an important parameter to consider if studying time of retrotransposition. We rewrote the paragraph to clarify the logic behind measurements of timing of GFP expression (subsection “LINE-1 retrotransposition peaks during S phase”).

12) How long does it take for a newly-integrated engineered L1 insertion to result in detectable GFP (or FT) expression? If TPRT occurs during S-phase, would the cell actually "turn green" during S-phase? The kinetics of doxycycline-induced GFP expression in Figure 5—figure supplement 1 suggest that it takes around 3 hours from the time reporter gene expression is induced to achieve detectable expression in 50% of cells, and 8 hours to get detectable expression in 90% of cells. It seems that a similar delay would apply to expression of a reporter gene delivered by L1 retrotransposition.

Yes! As mentioned in the previous response we measured GFP expression timing to try to quantify the time needed for a newly-integrated engineered L1 insertion to result in detectable GFP (or FT) expression. The mere fact we are able to observe blue-only cells demonstrates that translation and accumulation of protein from a retrotransposed FT gene occurred within less than 3 hrs. Our experiments using Tet-CMV-GFP expression show that transcription, translation and accumulation of GFP can occur in less than 3 hrs. As reported #11 above, to observe blue-only cells, use of a strong promoter such as CMV to drive FT is necessary and our measurements of FT expression timing indicate that transcription from CMV is fast enough to produce visible proteins <3 h. Our measurement upon nocodazole synchronization and release show that cells transition through S in about 5-6 h. Considering our new experiments (Figure 5) showing that increasing the time the cells spend in G1 (or G1/S boundary) does not increase retrotransposition (Figure 5), we conclude that retrotransposition and the events necessary for its detection (transcription, translation and accumulation of the fluorescent-AI reporter) can all occur in S phase.